# Prevention of dsRNA-induced interferon signaling by AGO1x is linked to breast cancer cell proliferation

Souvik Ghosh[1,†] [ID], Joao C Guimaraes[1,*,†] [ID], Manuela Lanzafame[2], Alexander Schmidt[3], Afzal Pasha Syed[1], Beatrice Dimitriades[1], Anastasiya Börsch[1], Shreemoyee Ghosh[1] [ID], Nitish Mittal[1], Thomas Montavon[4], Ana Luisa Correia[5], Johannes Danner[6], Gunter Meister[6], Luigi M Terracciano[2], Sébastien Pfeffer[4], Salvatore Piscuoglio[2,5] & Mihaela Zavolan[1,**] [ID]

## Abstract

Translational readthrough, i.e., elongation of polypeptide chains beyond the stop codon, was initially reported for viral RNA, but later found also on eukaryotic transcripts, resulting in proteome diversification and protein-level modulation. Here, we report that AGO1x, an evolutionarily conserved translational readthrough isoform of Argonaute 1, is generated in highly proliferative breast cancer cells, where it curbs accumulation of double-stranded RNAs (dsRNAs) and consequent induction of interferon responses and apoptosis. In contrast to other mammalian Argonaute protein family members with primarily cytoplasmic functions, AGO1x exhibits nuclear localization in the vicinity of nucleoli. We identify AGO1x interaction with the polyribonucleotide nucleotidyltransferase 1 (PNPT1) and show that the depletion of this protein further augments dsRNA accumulation. Our study thus uncovers a novel function of an Argonaute protein in buffering the endogenous dsRNA-induced interferon responses, different than the canonical function of AGO proteins in the miRNA effector pathway. As AGO1x expression is tightly linked to breast cancer cell proliferation, our study thus suggests a new direction for limiting tumor growth.

**Keywords** Argonaute 1; breast cancer; endogenous dsRNA; interferon response; translation readthrough
**Subject Categories** RNA Biology
**The EMBO Journal (2020) 39: e103922**

## Introduction

Guided by small RNAs—miRNA or siRNA—the four members of the human Argonaute (AGO) protein family repress translation and promote degradation of mRNA targets (Meister, 2013), with largely overlapping target specificities (Landthaler *et al*, 2008). AGO proteins load their small RNA partners from double-stranded (ds) precursor RNAs produced by the RNase III enzyme Dicer; typically, only one strand of the duplex, referred to as the guide strand, finds its way into the AGO protein (Kobayashi & Tomari, 2016). Mammalian AGO proteins are primarily found in the cytoplasm, and other patterns of cellular localization are debated (Bartel, 2018). However, Argonaute 1 (AGO1 or EIF2C1) has been detected at promoters and enhancers, modulating chromatin marks (Ameyar-Zazoua *et al*, 2012), transcription (Huang *et al*, 2013; Skourti-Stathaki *et al*, 2014), and alternative splicing (Ameyar-Zazoua *et al*, 2012; Alló *et al*, 2014). In mouse embryonic stem cells, human induced pluripotent stem cells, and mouse and human myoblasts, a large fraction of the Argonaute 2 was found in the nucleus, carrying out gene-silencing functions through CCR4/NOT-dependent mechanisms, similar to those described for cytoplasm-localized Argonaute proteins (Sarshad *et al*, 2018). Some studies indicate that the activity and subcellular localization of AGO proteins to processing (P) bodies, stress granules, or organelles such as mitochondria, endoplasmic reticulum (ER), and exosomes can be changed via post-translational modifications (Leung, 2015), but this aspect has not been extensively investigated. Ubiquitination remains the only discovered modification that leads to the degradation of AGO2 by the proteasome complex in a mammalian system (Smibert *et al*, 2013).

---

1 Computational and Systems Biology, Biozentrum, University of Basel, Basel, Switzerland
2 Institute of Pathology, University Hospital Basel, Basel, Switzerland
3 Proteomics Core Facility, Biozentrum, University of Basel, Basel, Switzerland
4 Architecture et Réactivité de l'ARN, Institut de biologie moléculaire et cellulaire du CNRS, Université de Strasbourg, Strasbourg, France
5 Department of Biomedicine, University of Basel/University Hospital Basel, Basel, Switzerland
6 Department of Biochemistry, Department of Biology and Preclinical Medicine, University of Regensburg, Regensburg, Germany
*Corresponding author. Tel: +41 61 682 28 97; E-mail: joaoguima@gmail.com
**Corresponding author. Tel: +41 61 207 15 77; E-mail: mihaela.zavolan@unibas.ch
†These authors contributed equally to this work

In the standard genetic code, TAA, TAG, and TGA are "stop" codons; they do not have a corresponding tRNA and are instead recognized by release factors, which triggers the termination of peptide synthesis and the release of ribosomes from the mRNA (Korostelev, 2011). Reliable recognition of stop codons is essential for protein synthesis; however, in some circumstances it has been found that the ribosome incorporates a standard or specialized amino acid at the stop codon and then continues to translate into the normally untranslated region (UTR) of the transcript up to the next stop codon (von der Haar & Tuite, 2007). This mechanism of stop codon or translation readthrough (TR) is exploited by retroviruses in the expression of their replication enzymes (Yoshinaka *et al*, 1985; Csibra *et al*, 2014). It has also been reported for some eukaryotic genes, especially since the development of transcriptome-wide ribosome occupancy profiling (Dunn *et al*, 2013; Dabrowski *et al*, 2015). By extending the C-terminus, TR can potentially impact the structure, function, and localization of proteins (Masel & Siegal, 2009). As TR has been specifically associated with stress (Gerashchenko *et al*, 2012), and many putative TR sites are evolutionarily conserved (Jungreis *et al*, 2016), TR may diversify the proteome and support the evolvability of protein sequences (Nelson & Masel, 2018).

In a study of programmed TR, *AGO1* emerged among the best predicted substrates (Eswarappa *et al*, 2014). Initially, the expression of AGO1x, the readthrough product of *AGO1,* from a reporter construct was demonstrated in HEK293 cells (Eswarappa *et al*, 2014), while a more recent study has reported expression of endogenous AGO1x in a variety of cells, including HEK293T and HeLa (Singh *et al*, 2019). Using an ectopic expression system, the latter study found that the let-7a miRNA promotes *AGO1* TR and that the cytoplasmically localized AGO1x has impaired gene-silencing function due to its inability to recruit GW182 to target mRNAs (Singh *et al*, 2019). It was thus concluded that AGO1x is a competitive inhibitor of the miRNA pathway. Whether the endogenous AGO1x behaves in a similar manner is still unclear.

In this study, we use multiple independent assays to demonstrate that AGO1x is expressed in human cell lines and tissues. By genetic depletion of AGO1x in two distinct cell lines, we show that AGO1x is involved in the response to endogenous double-stranded RNAs (dsRNAs), since its depletion leads to the accumulation of dsRNAs, activation of the interferon (IFN) response, and increased apoptosis. Pharmacological inhibition of Janus kinases (JAK) that are downstream of the IFN-α/β pathway is able to revert the AGO1x depletion phenotype, demonstrating that the interferon response is activated in AGO1x-deficient breast cancer cells. Consistently, AGO1x-deficient cells are more resistant to virus infection compared to wild-type cells. Our data further show that endogenous AGO1x localizes to the nucleus where it interacts with PNPT1, whose depletion further augments dsRNA accumulation. Finally, we find that AGO1x expression is increased in breast cancer patient samples along with expression of proliferation markers, which suggests that AGO1x may promote tumor growth.

## Results

### AGO1x is expressed in a variety of human cell lines

Evidence for the importance of translation in the region downstream of the canonical open reading frames can be obtained by analyzing the pattern of nucleotide conservation of these regions among vertebrates. Averaging per-nucleotide PhastCons conservation scores (Siepel *et al*, 2005) computed from alignments of 45 different vertebrate species (provided by the UCSC Genome Browser (Kent *et al*, 2002)) across regions downstream of canonical stop stop codons of RefSeq transcripts (NCBI Resource Coordinators, 2013), we found that only ~5% of the transcripts harbor highly conserved regions that may encode C-terminal protein extensions (Fig 1A). Interestingly, the proteins encoded by these transcripts tend to be RNA- and protein-interacting proteins, involved in the regulation of gene expression and metabolism. The *AGO1* transcript has been shown to be a good substrate for TR, but the function of the encoded protein has only started to be characterized (Eswarappa *et al*, 2014; Singh *et al*, 2019). All Argonaute family members exhibit high conservation downstream of their annotated stop codons, but the conservation is particularly striking for *AGO1* (Fig 1A). The multiple sequence alignment that covers the 99-nt-long region extending from the annotated stop codon of *AGO1* to the closest stop codon predicted further downstream is almost entirely devoid of nonsense mutations (Fig 1B). Furthermore, the alignment contains only a single deletion, in Tarsiers, and this deletion encompasses 3 nts, thus preserving the reading frame. This pattern of conservation strongly indicates that the region downstream of the canonical stop codon of *AGO1* is under selection for maintaining the reading frame. As expected from the genome alignment, the predicted C-terminal extension of the protein is also extremely conserved across vertebrates (Fig 1C).

To identify a cellular system in which to characterize the function of endogenous AGO1x, we sought evidence of its expression in a few model cell lines by Western blotting. While the level of the *AGO1* mRNA varies relatively little across normal tissues (Appendix Fig S1A) (Uhlén *et al*, 2015), a previous study reported an important role for AGO1 in breast cancer (Sung *et al*, 2011). Therefore, we included a few breast cancer cell lines in our analysis. Using a commercially available AGO1 antibody, we found a strong primary signal, as expected. In addition, we also consistently observed a cryptic second band of higher molecular weight, which had variable intensity across cell lines (Fig 1D top panel and Fig EV1A). Surmising that this band corresponds to AGO1x, we obtained a polyclonal antibody directed to the peptide predicted from the readthrough region (marked by the red line in Fig 1C). To thoroughly characterize the specificity of this antibody, we examined the expression of both AGO1 and AGO1x in the MDA-MB-231 breast cancer cell line upon perturbations. First, we transiently overexpressed FLAG-tagged variants of either AGO1 or AGO1x, mutating the canonical TGA stop codon to a serine-encoding TCC codon to generate the C-terminally extended isoform of AGO1 (AGO1x). We verified the overexpression of these proteins by Western blotting using a FLAG-specific antibody (Fig EV1B), which further revealed the difference in size between the AGO1 and AGO1x isoforms. While the commercial AGO1 antibody detected two bands, as before, the intensity of the bands increased in proportion to the expression of the FLAG-tagged constructs; the band corresponding to AGO1 had much higher intensity in cells transfected with *FLAG-AGO1,* whereas the band corresponding to AGO1x increased in intensity upon *FLAG-AGO1x* transfection (Fig EV1B, top panel). We also constructed two cell lines stably expressing either *FLAG-AGO1* or *FLAG-AGO1x* and carried out Western blotting using either the

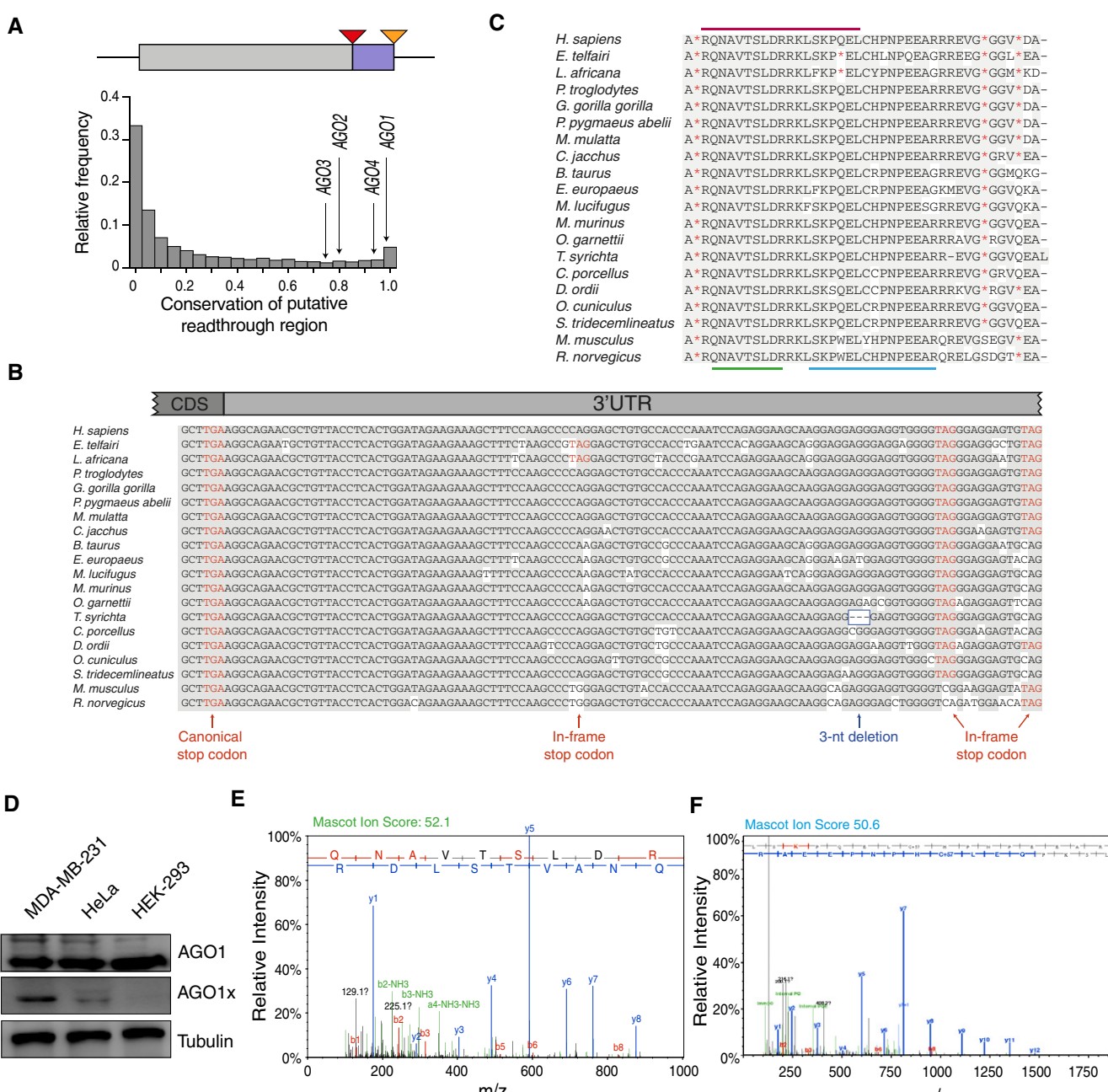

**Figure 1.   Evidence of *AGO1* transcript translational readthrough and of AGO1x expression.**

A   Top: schema of analyzed TR regions (purple), located downstream of the annotated open reading frame (gray), between the annotated stop codon (red triangle) and the next in-frame stop codon (orange triangle); Bottom: histogram of average PhastCons conservation scores (*x*-axis) of putative TR regions of all RefSeq-annotated transcripts. The scores of the four human Argonaute protein family members are highlighted.

B   Multiple sequence alignment of the *AGO1* putative TR region across vertebrates.

C   Multiple sequence alignment of the corresponding predicted amino acid sequence. The unique peptide targeted by the polyclonal antibody is indicated by the red line. The green and blue lines indicate peptide sequences obtained after tryptic digestion, in which cleavage is exclusively at arginine (R) and lysine (K) (further described below in panels E and F). Red asterisks indicate stop codons.

D   Western blot showing AGO1x expression in three cell lines. For comparison, a parallel blot was probed with an antibody directed against canonical AGO1. Tubulin served as loading control.

E, F   Annotated MS/MS spectrum of peptides specific for the endogenous AGO1x, "QNAVTSLDR", depicted in green (E) and "LSKPQELCHPNPEEAR", depicted in blue (F). The Mascot ion score (text color corresponds to peptides marked in Fig 1C for reference) as well as the annotated fragments (blue = y-ions; red = b-ions) together with the corresponding amino acids is indicated.

FLAG or the AGO1x antibody. These experiments again demonstrated that the AGO1x antibody is highly specific for AGO1x (Fig EV1C). Further cellular fractionation showed that the overexpressed AGO1 and AGO1x localize to the cytoplasm (Fig EV1D). As AGO1 and AGO1x are translation-generated isoforms encoded by the same transcript, they cannot be depleted independently by means of siRNAs. Nevertheless, treatment with an siRNA pool designed for the *AGO1* transcript elicited the expected effects of both AGO1 and AGO1x. Namely, in MDA-MB-231 cells with endogenous expression of AGO1 and AGO1x, the commercial AGO1 antibody gave both cytoplasmic and nuclear signals, both of which were reduced upon siRNA treatment (Fig EV1E, left-most panels). In contrast, the AGO1x-specific antibody revealed only a nuclear signal, also reduced upon the siRNA-mediated depletion of *AGO1* (Fig EV1E). In the cells expressing *FLAG-AGO1x*, both the FLAG and the commercial AGO1 antibody detected the previously observed cytoplasmic signal, which was strongly reduced by the siRNA treatment (Fig EV1F). Similarly, the AGO1x antibody detected both cytoplasmic and nuclear signals, which were reduced by the siRNA treatment. Finally, we re-evaluated the nature of the second band we observed in WBs carried out with the commercial AGO1 antibody (Fig 1D top panel) by probing a parallel blot from the same cell lysates with the AGO1x antibody, confirming that the band corresponds to AGO1x (Fig 1D middle panel). Altogether, these results indicate that our AGO1x antibody specifically recognizes the AGO1x isoform.

The findings that endogenously expressed AGO1x localizes primarily to the nucleus of MDA-MB-231 cells, whereas the FLAG-tagged variant remains mostly in the cytoplasm (Fig EV1D–F) were unexpected. While we did not find the reason behind this localization pattern, ectopically expressed AGO1x was also found in the cytoplasm in a previous study (Singh *et al*, 2019). Thus, we decided to focus on cellular systems with endogenous AGO1x expression (Figs 1D and EV1A, Appendix Fig S2A). We further used the cell lines that showed the highest AGO1x expression in our analysis, which were the MDA-MB-231, a cell line derived from the metastatic site of a human breast tumor, and HeLa, a cell line derived from a malignant cervical tumor (Figs 1D and EV1A). We did not use HEK 293 lines [HEK293FT (Fig 1D) and HEK293T cells (Appendix Fig S2B)], where the expression of AGO1x was less clear. We noted also that the relative abundances of AGO1x in the cell lines that we studied matched the relative abundance of the let-7a miRNA (Appendix Fig S2C), in agreement with the finding that let-7a drives the TR of *AGO1* (Singh *et al*, 2019).

To obtain independent evidence that AGO1x is present in MDA-MB-231 cells, we carried out targeted LC-MS analysis and identified two peptides that are predicted from the readthrough region, both with very high confidence [FDR < 1% based on a target-decoy search strategy (Elias & Gygi, 2007)]. These peptides, marked by the green and blue lines in Fig 1C, are QNAVTSLDR and LSKPQELCHPNPEEAR and had Mascot ion scores of 52.1 and 50.6, respectively (Fig 1E and F). The joint identification of these peptides, never observed in previous analyses (Singh *et al*, 2019), provides strong evidence for AGO1x expression. To further exclude that these peptides can be derived from another protein, we searched the entire reference human proteome by translated BLAST (Madden, 2013) using the NCBI server. The search did not reveal any hits beyond the expected *AGO1* locus. Together, the

perturbation experiments, Western blotting, and proteomic analyses demonstrate that the AGO1x isoform is expressed in human breast cancer cell lines, in all likelihood by TR.

### Endogenous AGO1x localizes primarily to the nucleus

Following the observations we made when characterizing the AGO1x antibody, we next sought to more precisely determine the localization of the protein by co-immunofluorescence with several nuclear and cytoplasmic markers. We used antibodies for SC35, Lsm4, alpha-tubulin, ERP72, p54(NRB)/NONO, and nucleolin to mark, respectively, nuclear speckles (Fei *et al*, 2017), cytoplasmic P bodies (Decker *et al*, 2007), cytoplasmic microtubules (Brinkley *et al*, 1975), endoplasmic reticulum (Mazzarella *et al*, 1990), nuclear paraspeckles (Fox *et al*, 2005), and nucleoli (Ma *et al*, 2007). This revealed the enrichment of AGO1x in the nuclear compartment of both MDA-MB-231 (Fig 2A) and HeLa (Appendix Fig S3A) cells, with little signal from the cytoplasm. Furthermore, within the nuclear compartment, AGO1x appeared enriched around nucleoli (Fig 2A, bottom right panel). We verified this partial co-localization by analyzing image stacks (Appendix Fig S3B and C) which showed a ~3-fold higher AGO1x signal intensity in nucleoli (defined based on the nucleolin signal) compared to nucleoplasm (Fig 2B). Western blots from nuclear and cytoplasmic fractions of MDA-MB-231 cells corroborated our initial observation that AGO1x is located primarily in the nucleus while the lower molecular weight and more abundant canonical isoform is essentially cytoplasmic (Fig 2C).

### Specific depletion of AGO1x leads to growth defects

To determine the consequences of specific depletion of AGO1x, we next used the CRISPR/Cas9 genome editing tool to induce frame-shifting mutations in the TR region. We used two parental cell lines, MDA-MB-231 and HeLa, to verify the generality of our findings. To control for CRISPR/Cas9 treatment effects that are not specifically due to AGO1x, we generated control lines, by treating the parental cells with Cas9 and a control single-guide RNAs (sgRNA) designed to target GFP, which is not expressed in our system. By using two sgRNAs directed against distinct sites of the TR region (Fig 3A, top panel), we derived two independent AGO1x-deficient lines from each of the parental lines. As the two sgRNAs have distinct spectra of putative off-target effects, they provide a control for potential off-target effects of the treatment. We selected individual clones in which the CRISPR/Cas9-induced DNA break was repaired by non-homologous end joining, resulting in out-of-frame deletion mutations in the TR region (Fig 3A, top panel, and Appendix Fig S4A and B). We have further sequenced the RNAs from our cell lines (Appendix Fig S4C and D). The alignment of the reads confirmed the deletions in the TR region, but some wild-type sequences were present as well (Appendix Fig S4C and D), indicating that not all genomic copies of *AGO1* underwent editing. We noted also that the sequencing data did not contain a single read supporting the scenario that the C-terminal extension of AGO1 can be generated from an alternative transcript, in which the annotated stop codon of *AGO1* is bypassed by alternative splicing.

Western blotting showed the successful depletion of AGO1x protein in both mutant MDA-MB-231 cell lines, with no apparent

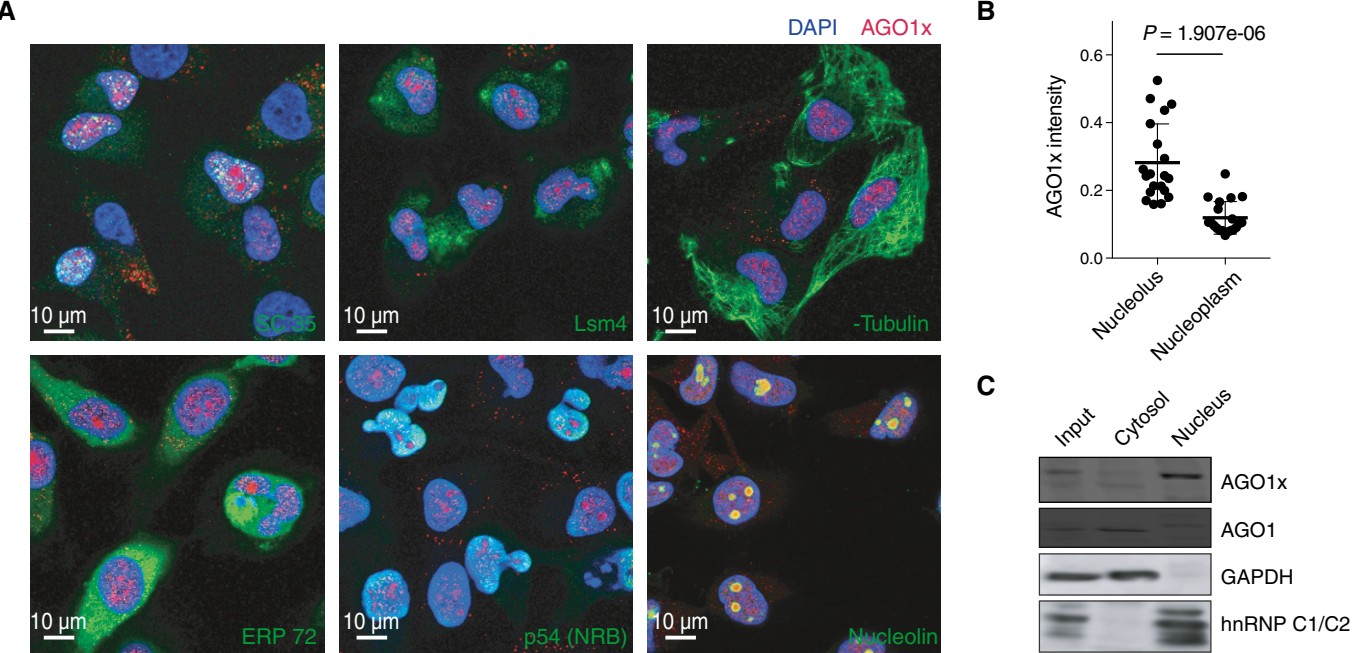

**Figure 2. AGO1x localizes to the nuclear region.**

A   Representative immunofluorescence images showing the subcellular distribution of AGO1x (red) relative to nuclear and cytoplasmic markers. DAPI was used to mark the nucleus (blue). The co-stained subcellular marker is indicated in each panel in green. SC 35, Lsm4, α-tubulin, ERP72, p54 (NRB), and nucleolin serve as markers for nuclear speckles, cytosol and nucleus, cytosol, endoplasmic reticulum, paraspeckle, and nucleolus, respectively.

B   Mean (± SD) pixel intensities of AGO1x staining in nucleolus and nucleoplasm, computed from z-stack images of MDA-MB-231 cells (*n* = 20). The *P*-value was determined using a paired two-tailed *t*-test.

C   Representative AGO1x and AGO1 blot from MDA-MB-231 cell fractions. GAPDH and hnRNP C1/C2 served as markers of purity of the individual fractions.

Source data are available online for this figure.

effect on the total AGO1 levels (Fig 3A, bottom panel). To resolve AGO1 and AGO1x on the same gel, lysates from control and mutant MDA-MB-231 cells were blotted from 10% SDS–PAGE gels on which electrophoresis was carried out for a protracted duration. The intensity of the higher molecular weight (AGO1x) band that was observed in addition to the canonical AGO1 isoform when probed with the AGO1 antibody was strongly reduced in the mutant lines (Appendix Fig S4E). Probing these gels with the AGO1x antibody confirmed this behavior, further reflected in the quantification of these bands (Appendix Fig S4F). In culture, mutant cell lines exhibited a marked reduction in growth relative to the parental line, documented first by microscopy at two different time points after seeding equal numbers of cells from the different lines (Fig 3B) and then with a non-invasive, electrical impedance-based method for quantifying real-time cell growth (Atienza *et al*, 2005) (Fig 3C). Mutant HeLa lines generated by CRISPR/Cas9 genome editing with the same sgRNAs exhibited similar growth defects (Appendix Fig S5A–C). These results demonstrate that the depletion of AGO1x isoform by sgRNA-mediated genome editing of *AGO1* readthrough region leads to growth defects. We also assessed the migration potential of cells by growing them in chambers with gold microelectrodes embedded in the bottom of microtiter wells for non-invasive electronic monitoring. These measurements indicated that beside slower growth, AGO1x-deficient cell lines also had a reduced migration potential (Fig 3D). Altogether, these assays suggested that AGO1x sustains

cell growth, which may be linked to its expression in the context of cancer cell lines.

To further test this hypothesis, we evaluated the impact of AGO1x depletion on two tumor-promoting properties, namely anchorage-independent growth, assessed by soft agar colony assay (Roberts *et al*, 1985), and ability to self-renew *in vitro*, assessed by the sphere formation assay (Ponti *et al*, 2005). We found that both mutant MDA-MB-231 cell lines showed strong impairment in these assays relative to the control line (Figs 3E and F, EV2A and B, respectively), consistent with the notion that AGO1x supports the development of breast cancer. To further validate this hypothesis, we examined patient-derived, paraffin-embedded breast cancer tissue sections by immunohistochemistry, using both the AGO1x antibody and an antibody for the proliferation marker Ki-67 (Li *et al*, 2015). Classifying the tumors into lowly or highly proliferative based on Ki-67-positive cells in accordance with the St. Gallen's guidelines (Piscuoglio *et al*, 2016), we strikingly found that not only was AGO1x present in these samples (Fig 3G), but also that its expression was strongly correlated with the proliferative index of the tumors (Fig 3G and H). Specifically, while in the Ki-67$^{low}$ only ~5% of cells were AGO1x-positive, ~65% of cells showed AGO1x positivity in the Ki-67$^{high}$ tumors. We further noted that the nuclear localization of the protein previously observed in cultured cells (Fig 2A) was also apparent in the patient samples (Fig 3G).

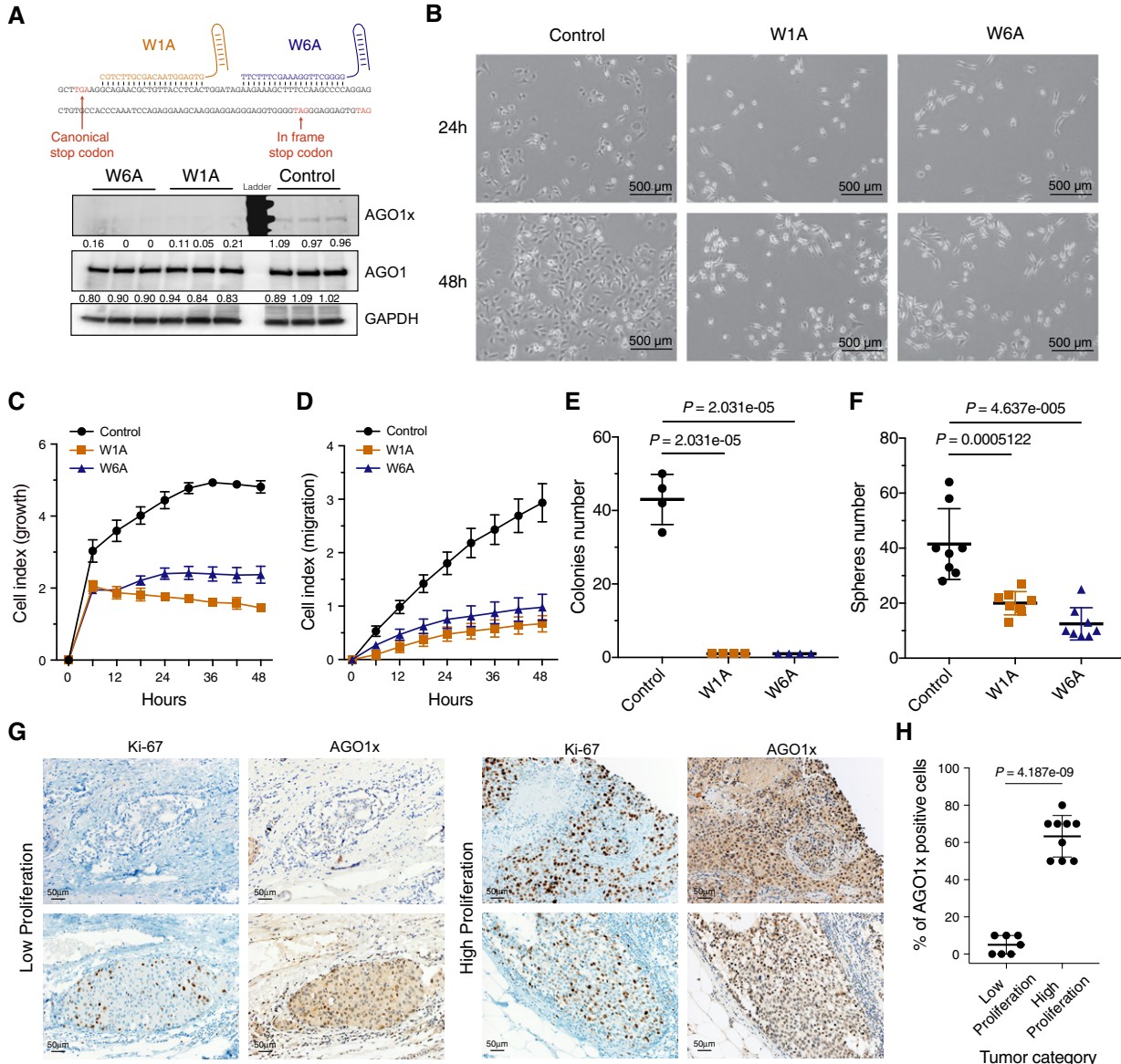

**Figure 3. AGO1x promotes cell proliferation.**

A Top: scheme of the sites targeted by sgRNA in the generation of AGO1x mutant cell lines. Bottom: Western blot analysis of lysates from mutant MDA-MB-231 cells showing the depletion of AGO1x and the comparable total AGO1 levels across the cell lines. Three independently collected sets of cells were lysed and analyzed in parallel. After calculating the levels of AGO1x and AGO1 relative to GAPDH, values corresponding to individual lanes were normalized to the average over the replicates of the control line. These numbers are indicated below the blots. The *P*-values of the *t*-tests comparing the expression AGO1x in mutant (W1A and W6A) and control (CTRL) lines were W1A-CTRL: 0.0008 and W6A-CTRL: 0.0002; and for AGO1, W1A-CTRL: 0.1227 and W6A-CTRL: 0.1173.

B Phase-contrast images of control and mutant cell lines at 24 and 48 h after plating equal numbers (10,000) of cells in individual wells of a six-well plate.

C Impedance-based mean (± SD) cell indices at the indicated time points after seeding equal numbers of CTRL (*n* = 6), W1A (*n* = 5), and W6A (*n* = 5) cells. From 6 h on, there is a statistically significant difference between control and the two mutant cell lines (*P* < 0.001, two-tailed *t*-test).

D Impedance-based mean (± SD) cell indices as a function of time for control (*n* = 3), W1A (*n* = 4), and W6A (*n* = 3) cells grown in electronically monitored Boyden chambers. From 12 h on, there is a statistical significant difference between control and the two mutants (*P* < 0.005, two-tailed *t*-test).

E Mean (± SD) number of colonies obtained in a soft agar colony formation assay of control, W1A, and W6A mutant cell lines (all *n* = 4, *P*-value determined using an unpaired two-tailed *t*-test).

F Mean (± SD) number of spheres obtained from a sphere formation assay of control, W1A, and W6A mutant cell lines (all *n* = 8, *P*-values computed as in E).

G AGO1x staining of tissue sections from breast cancers, classified as low proliferating or high proliferating based on Ki-67 staining. The upper and lower panel represents two different tissue sections.

H Mean (± SD) percentage of AGO1x-positive cells in breast tumors with low (*n* = 7) and high (*n* = 9) proliferation index, assessed by the expression level of Ki-67. *P*-value was determined using an unpaired two-tailed *t*-test.

Source data are available online for this figure.

### Interferon response pathways are activated in AGO1x-deficient lines

To better understand the consequences of AGO1x deficiency, we carried out RNA sequencing of both parental and mutant cell lines. Analysis of the RNA sequencing data obtained from the control and the two mutant MDA-MB-231 cell lines revealed that hundreds of genes had significantly different expression levels (|fold change| > 2-fold and FDR < 0.01, Appendix Fig S6A and B) in mutant cells

compared to control. Importantly, the differences in gene expression between the mutant lines generated using different sgRNAs and the control cells were highly correlated ($R = 0.92$, $P < 0.001$, Fig 4A). This points to the targeting of *AGO1x* being at the origin of these differences and not an off-target effect of sgRNAs, which should be sgRNA sequence-specific and thereby different between the mutant lines. Gene set enrichment analysis (GSEA) (Subramanian *et al*, 2007) showed that genes from the interferon-alpha response and apoptosis pathways had increased expression levels in mutant cell

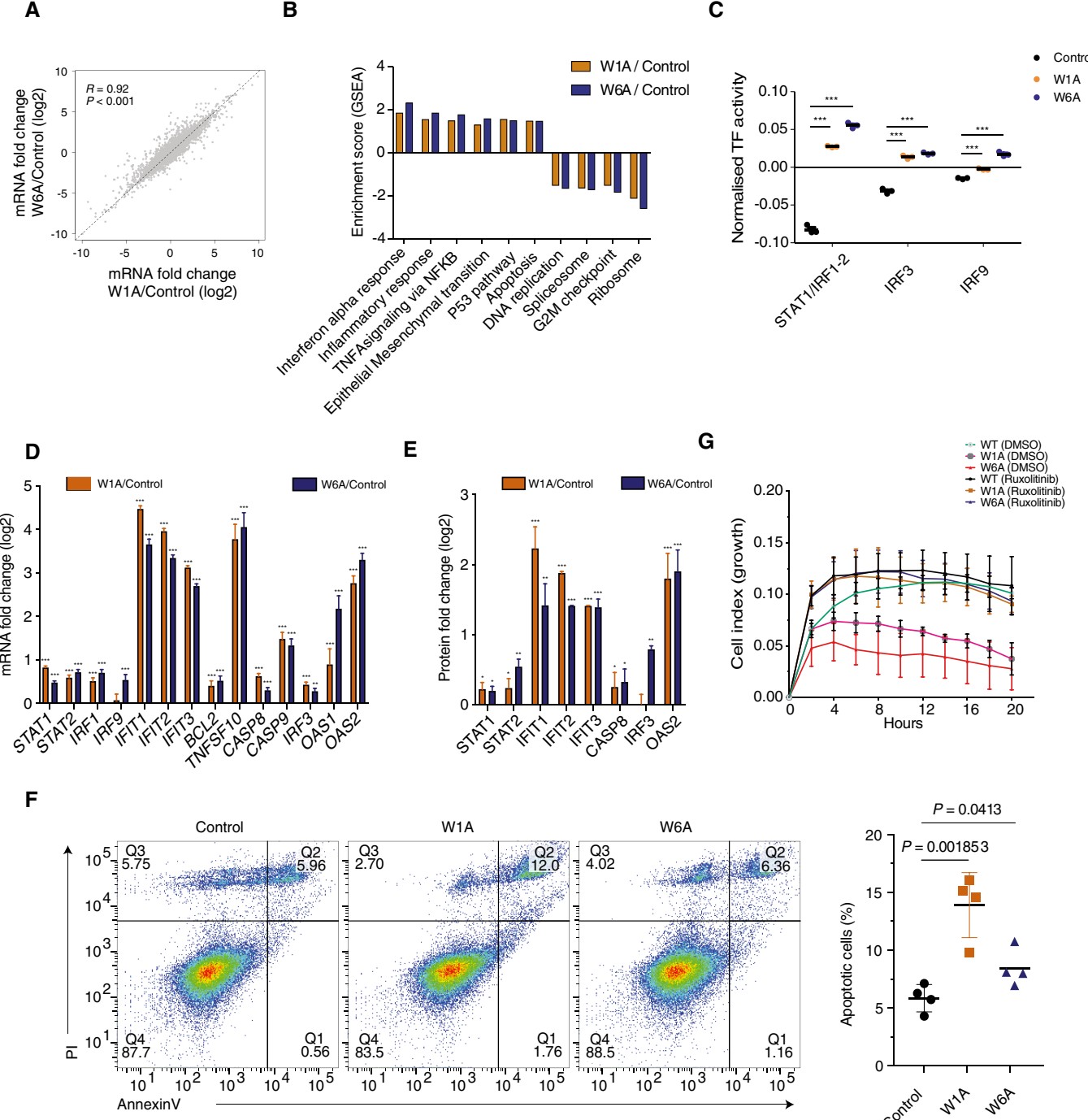

**Figure 4.**

◀

**Figure 4.   Depletion of AGO1x leads to increased interferon response and apoptosis.**

A   Scatter plot of mean log2 mRNA fold changes in the mutant cell lines versus control ($n = 3$). Shown are also the Pearson correlation coefficient and respective *P*-value. The dashed line indicates equal change in the two mutant lines.

B   Normalized enrichment score (ES) from gene set enrichment analysis (Subramanian *et al*, 2005) comparing gene expression changes in the mutant cell lines relative to the control cell line. For all depicted pathways, $P < 0.05$. *P*-values were calculated by comparing the empirical ES of a gene set to a null distribution of ESs derived from permuting the gene set and then adjusted for multiple hypothesis testing.

C   Mean ($\pm$ SD) activity of STAT1/IRF1/IRF2/IRF3/IRF9 transcription factor motifs estimated by ISMARA (Balwierz *et al*, 2014) ($n = 3$). Shown are the *P*-values (***$P < 0.001$) determined by Dunn's multiple comparison test post hoc and the non-parametric Kruskal–Wallis test ($P < 0.0001$).

D   Mean ($\pm$ SEM) log2 mRNA expression fold changes (inferred from RNA-seq) of genes involved in the interferon-alpha response and apoptosis in the two mutant cell lines relative to control ($n = 3$). Multiple testing-corrected *P*-values for Wald tests comparing fold changes with respect to control are depicted above each bar (**$P < 0.01$, ***$P < 0.001$).

E   Mean ($\pm$ SD) expression fold changes of the corresponding proteins (if detected in the proteomics data) in the two mutant cell lines (W1A, $n = 2$; W6A, $n = 1$) compared to control ($n = 2$). Multiple testing-corrected *P*-values for unpaired two-tailed *t*-tests (proteomics) comparing fold changes with respect to control are depicted above each bar (*$P < 0.05$, **$P < 0.01$, ***$P < 0.001$).

F   Representative result of the apoptosis assay using annexin V and propidium iodide (PI) staining in the control and the two mutant cell lines (left). The percentage of cells in each quadrant is depicted for each cell line. Quantification of the mean ($\pm$ SD) percentage of apoptotic cells (Q1+Q2, Annexin V+) across the different cell lines ($n = 4$) (right). Shown is the *P*-value determined by unpaired two-tailed *t*-test.

G   Impedance-based mean ($\pm$ SD) cell index values at the indicated time points of growth after seeding equal numbers of control, W1A, and W6A cells, upon treatment with DMSO or ruxolitinib. For the DMSO treatment: control ($n = 4$), W1A ($n = 3$), and W6A ($n = 3$). For the ruxolitinib treatment: control ($n = 6$), W1A ($n = 3$), and W6A ($n = 4$). From 6 h after DMSO treatment, there is a statistically significant difference between control and the two mutants ($P < 0.05$, two-tailed *t*-test), whereas no statistically significant difference is found after ruxolitinib treatment.

Source data are available online for this figure.

lines relative to control (Fig 4B). In contrast, genes involved in cell cycle progression and those encoding ribosomal proteins had decreased expression in the mutant lines (Fig 4B).

To uncover the transcriptional networks underlying these gene expression changes, we used the ISMARA tool (Balwierz *et al*, 2014), which identifies transcription factors and miRNAs that best explain measured changes in mRNA expression. ISMARA predicted an increased activity of STAT1 and IRF1/2/3/9 transcription factors in the mutant cell lines (Fig 4C), implicating the signaling through IFN-α/β receptors in the observed gene expression changes. Interferons have several important biological activities, including antiviral, immunomodulatory, antiproliferative, and pro-apoptotic (Bekisz *et al*, 2010; Schneider *et al*, 2014). Multiple genes from the interferon pathway (Schneider *et al*, 2014) were upregulated at the mRNA level (shown by the RNA-seq data, Fig 4D). We further carried out a proteome analysis by TMT-based protein labeling followed by LC-MS/MS (Fig 4E). Although the coverage of the proteome was more limited than the coverage of the transcriptome, interferon-induced proteins also showed increased expression in the mutant lines. Supporting the GSEA-inferred upregulation of the apoptosis pathway (Fig 4B), caspases 8 and 9 (Balachandran *et al*, 2000) and the tumor necrosis factor-related apoptosis-inducing ligand TNFSF10/TRAIL were increased significantly at RNA level, though they were poorly detected in the proteome analysis. Thus, to further clarify whether apoptosis is increased in the mutant lines, we performed immunostaining of propidium iodide (PI) and annexin V, measuring the steady-state abundance of apoptotic cells. Indeed, this analysis showed that apoptosis was increased in the MDA-MB-231 mutant cell lines compared to the control (Fig 4F). We also measured directly the CASP3/8/9 activity and the extent of PARP cleavage and found that they were both enhanced upon depletion of AGO1x in the mutant cell lines (Appendix Fig S7A–D).

To further validate that the growth defect of the mutant lines was due to the activation of the interferon pathway, we carried out a growth assay in the absence or presence of ruxolitinib, a JAK1 inhibitor that specifically blocks the interferon response. We found that this treatment rescued the cell growth (Fig 4G), demonstrating that

the activation of the interferon response following AGO1x depletion accounted for the increased apoptosis and growth defect phenotypes. We repeated the RNA-seq profiling and analyses in HeLa cell lines in which the AGO1x readthrough region was similarly targeted genetically and found a similar upregulation of interferon response and apoptosis pathways (Fig EV3A–G). As HeLa cells are susceptible to Sindbis virus (SINV), we also used a virus infection assay for a final readout of the increased interferon pathway activity. Indeed, we found that SINV accumulation was lower 48 h post-infection in HeLa mutant cell lines compared to the control as assessed by microscopy and Western blot analysis (Fig EV4A and B) and viral titer determination by plaque assay (Fig EV4C). These results demonstrate that the depletion of AGO1x leads to an increased activity of the interferon pathway, which has functional consequences for cell viability, growth, and infectivity by viruses.

**AGO1x-deficient cells accumulate dsRNAs**

Activation of the interferon response could be due to various causes, including the accumulation of dsRNAs (Field *et al*, 1967; Lampson *et al*, 1967) (Fig 5A). Thus, we next examined the levels of dsRNA sensors (Estornes *et al*, 2013; Schneider *et al*, 2014) in our cell lines. Expression of Toll-like receptor 3 (TLR3), a membrane receptor involved in the recognition of long (> 50 nucleotides) dsRNA intermediates of viral replication (Estornes *et al*, 2013), was largely unchanged in the mutant lines (Fig 5A and B). In contrast, the expression of RNA helicase RIG-I (also known as DDX58) and MDA-5 (melanoma differentiation-associated gene 5, also known as interferon induced with helicase C domain 1 or IFIH1) (Estornes *et al*, 2013), which recognize short and long cytoplasmic dsRNAs, respectively, was significantly increased at the RNA level (Fig 5B). DDX58 also increased significantly at the protein level, whereas IFIH1 was not detected in the proteome analysis (Fig 5C). Also induced in the mutant lines was the RNA-dependent serine/threonine kinase PKR (encoded by the eukaryotic translation initiation factor 2-alpha kinase 2 gene, EIF2AK2) (Fig 5B and C). Binding of dsRNAs to PKR promotes autophosphorylation, dimerization, and functional

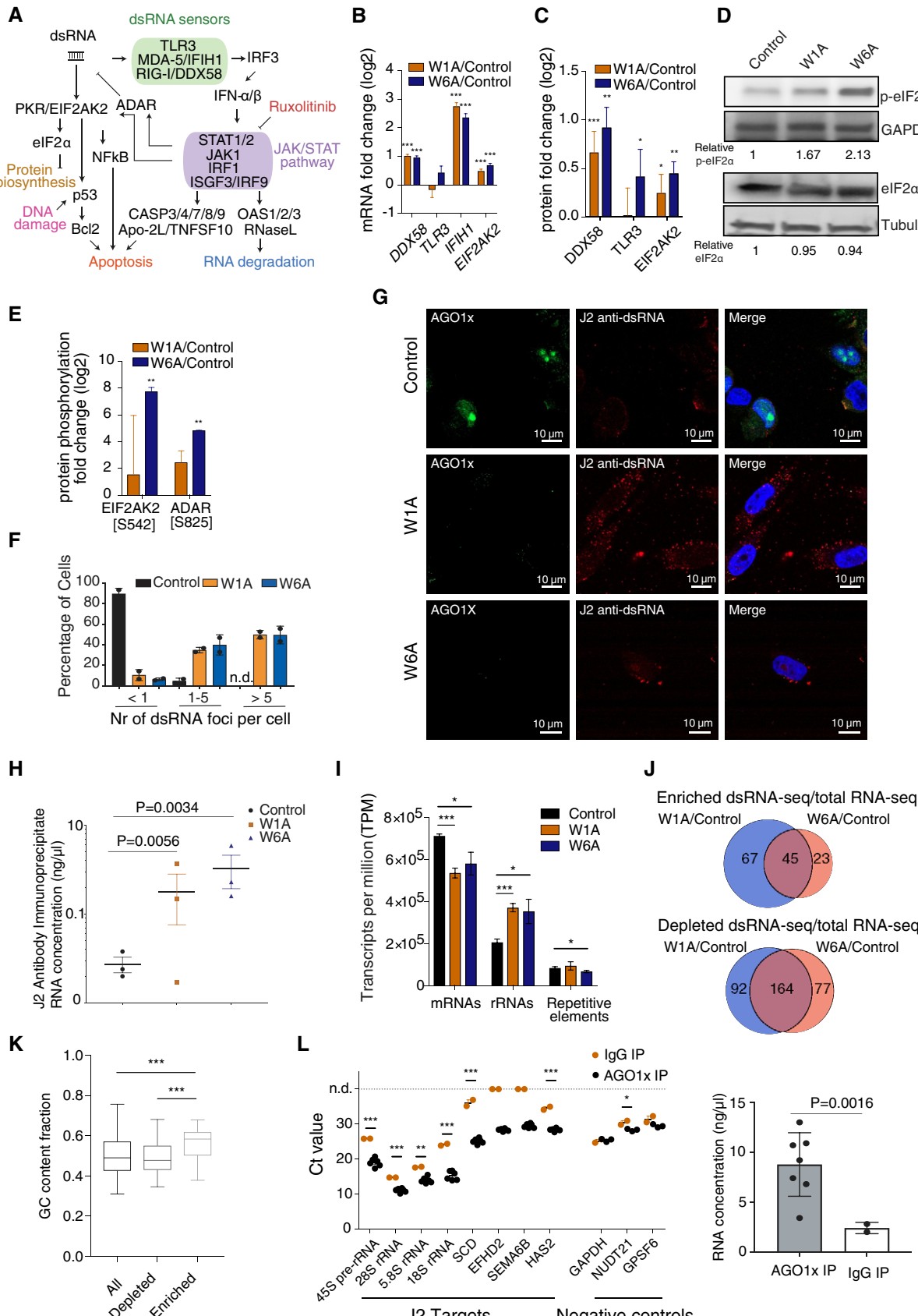

**Figure 5.**

**Figure 5.  Loss of AGO1x triggers accumulation of endogenous dsRNA.**

A Schematic representation of the dsRNA sensing mechanism, downstream signaling events, and expected phenotypes.

B Mean ($\pm$ SEM) log2 mRNA fold changes of dsRNAs sensors (from panel A) in the two mutant cell lines compared to control ($n$ = 3) inferred from RNA-seq. Multiple testing-corrected $P$-values for Wald tests comparing fold changes with respect to control are depicted above each bar (\*\*\*$P$ < 0.001).

C Mean ($\pm$ SD) fold changes of corresponding proteins (see B), if detected in the proteomics data from the two mutant (W1A, $n$ = 2; W6A, $n$ = 1) lines relative to control ($n$ = 2). Multiple testing-corrected $P$-values for unpaired two-tailed $t$-tests (proteomics) comparing fold changes with respect to control are depicted above each bar (\*$P$ < 0.05, \*\*$P$ < 0.01, \*\*\*$P$ < 0.001).

D Western blot detection of phosphorylated eIF2$\alpha$ (Ser51) and total eIF2$\alpha$ levels in cell lysates of control and mutant MDA-MB-231 cells. GAPDH and tubulin serve as loading control for phosphorylated eIF2$\alpha$ and total eIF2$\alpha$ blots, respectively. Numbers indicated below the blots represent the band intensities of target proteins normalized to the loading controls, with the value in the control line taken as baseline of 1.

E Mean ($\pm$ SD) fold changes in phosphorylated peptide (phosphorylation sites are indicated in brackets) levels derived from a proteomic analysis of phosphopeptides in the two mutant lines (W1A, $n$ = 2; W6A, $n$ = 2) compared to control ($n$ = 1). Multiple testing-corrected $P$-values for Wald tests (transcriptomics) or unpaired two-tailed $t$-tests (proteomics) comparing fold changes with respect to control are depicted above each bar (\*\*$P$ < 0.01).

F Quantification of the relative abundance (mean $\pm$ SEM) of distinct dsRNA foci in each cell type. 3D images were projected on a single plane, non-overlapping foci were counted from 100 cells of each type, and the numbers in three abundance bins (< 1, 1–5, > 5) are shown.

G Representative immunofluorescence image of the control and mutant MDA-MB-231 cells stained with AGO1x (green) and J2 antibody (red) and DAPI (blue) to mark the nucleus. Loss of AGO1x signal in mutants also confirms the efficacy of CRISPR-induced mutations.

H, I Concentration of recovered RNA (mean $\pm$ SEM) measured by NanoDrop™ (H) and relative abundance of various RNA species (mean $\pm$ SD) in J2 antibody immunoprecipitates from three biological replicates of control and mutant cell lines (I). Multiple testing-corrected $P$-values from unpaired two-tailed $t$-tests are depicted above each comparison (\*$P$ < 0.05, \*\*\*$P$ < 0.001).

J Venn diagrams showing the intersection between RNA transcripts that were consistently enriched (top) or depleted (bottom) in the dsRNA-seq relative to total RNA-seq in the two mutant cell lines compared to control ($n$ = 3).

K Boxplots showing the proportion of G/C nucleotides in all genes and in genes depleted/enriched in dsRNA-seq compared to total RNA-seq ($n$ = 3). Shown are the $P$-values in the non-parametric Mann–Whitney $U$ test (\*\*\*$P$ < 0.001). Boxes extend from the 25th to 75th percentiles (inter-quartile range (IQR)), horizontal lines represent the median, whiskers indicate the lowest and highest data within 1.5×IQR from the lower and upper quartiles, respectively.

L Left: quantification of transcript abundance (mean $\pm$ SEM) by qRT–PCR in AGO1x (black, $n$ = 6 for putative targets, $n$ = 3 for negative controls) or IgG (orange, $n$ = 2) IP from control cells. n.d., not detected. Multiple testing-corrected $P$-values for an unpaired two-tailed $t$-test are depicted above each comparison (\*$P$ < 0.05, \*\*$P$ < 0.01, \*\*\*$P$ < 0.001). Right: concentration of RNA (mean $\pm$ SD) obtained with each type of antibody, with $P$-value from an unpaired two-tailed Student's $t$-test.

Source data are available online for this figure.

activation of this protein (Dey *et al*, 2005). PKR is central to the response to dsRNAs (Fig 5A), as it phosphorylates the $\alpha$-subunit of eukaryotic initiation factor 2 to repress translation (Farrell *et al*, 1978; Lee & Esteban, 1993) and mediate apoptosis (Gil *et al*, 1999; Donzé *et al*, 2004; Hsu *et al*, 2004). Consistently, eIF2$\alpha$ phosphorylation was increased in the mutant lines relative to control, while the total eIF2$\alpha$ levels remained unchanged (Fig 5D and Appendix Fig S7D). Furthermore, by quantifying the relative abundance of phosphorylated peptides with a previously published method (Post *et al*, 2017) we identified increased phosphorylation at S542 in PKR and at the known S825 site (Sakurai *et al*, 2017) in the adenosine deaminase (ADAR), also involved in the response to dsRNAs (Fig 5E).

Finally, to directly evaluate the dsRNA levels in the different cell lines, we used the SCICONS J2-specific antibody, validated in a previous study (Schönborn *et al*, 1991). Staining of AGO1x mutant cells with this antibody indeed confirmed the accumulation of dsRNAs (Fig 5F and G). To determine their molecular origin, we sequenced the RNAs isolated by J2 antibody-based pulldown and found a very heterogeneous population, in which rRNA-derived sequences were strongly enriched (Fig 5H–J). Overall, both the rRNAs and the mRNAs that accumulated in the mutant lines had a higher proportion of G/C nucleotides and thus a higher propensity of forming stable secondary structures in comparison with RNAs that were not pulled down by the J2 antibody (Fig 5K). To confirm the direct interaction of AGO1x with RNAs that have increased propensity of forming base-paired structures (which, for simplicity, we call dsRNAs) and can be isolated with the J2 antibody from the mutant lines, we next immunoprecipitated AGO1x and carried out quantitative PCR with specific probes. We found that these RNAs

are also enriched in AGO1x IP, whereas control RNAs are not (Fig 5L). These analyses show that the loss of AGO1x leads to the accumulation of RNAs that have a high propensity to form stable structures, are recognized by the J2 antibody, and are also enriched in AGO1x pulldowns. The presence of these RNAs is correlated with activation of the interferon response.

## PNPT1 interacts specifically with AGO1x

To identify additional components of the dsRNA-dependent response in our cell lines, we analyzed the proteins that co-immunoprecipitated with AGO1x from nuclear fractions by mass spectrometry. Relative to the control IgG pulldown, very few proteins were significantly enriched by the AGO1x antibody. The top and most specific AGO1x interactor was the polyribonucleotide nucleotidyltransferase 1 (PNPT1), but a few others were identified, including DExH-Box helicase 9 (DHX9) and fibrillarin (FBL) (Fig 6A). We further confirmed the interaction of AGO1x with PNPT1 by IP with AGO1x-specific antibody in comparison with IgG, followed by Western blotting (Fig 6B) as well as by reciprocal IP of PNPT1, which identified AGO1x as interactor (Fig 6C and D). Additional *in vitro* interaction assays of FLAG-tagged AGO1 and AGO1x with nuclear extracts confirmed that AGO1x, but not AGO1, specifically interacts with PNPT1 (Fig 6E). PNPT1 is a bifunctional enzyme increasingly implicated in RNA metabolic processes, including the translocation of RNAs from the cytoplasm to the mitochondria (Wang *et al*, 2009), and the degradation of cytoplasmic c-MYC-encoding mRNA (Sarkar *et al*, 2003) and miR-221 microRNA (Das *et al*, 2010). Immunofluorescence analysis of PNPT1 in MDA-MB-231 further revealed the overlap of PNPT1 signal from the nucleus

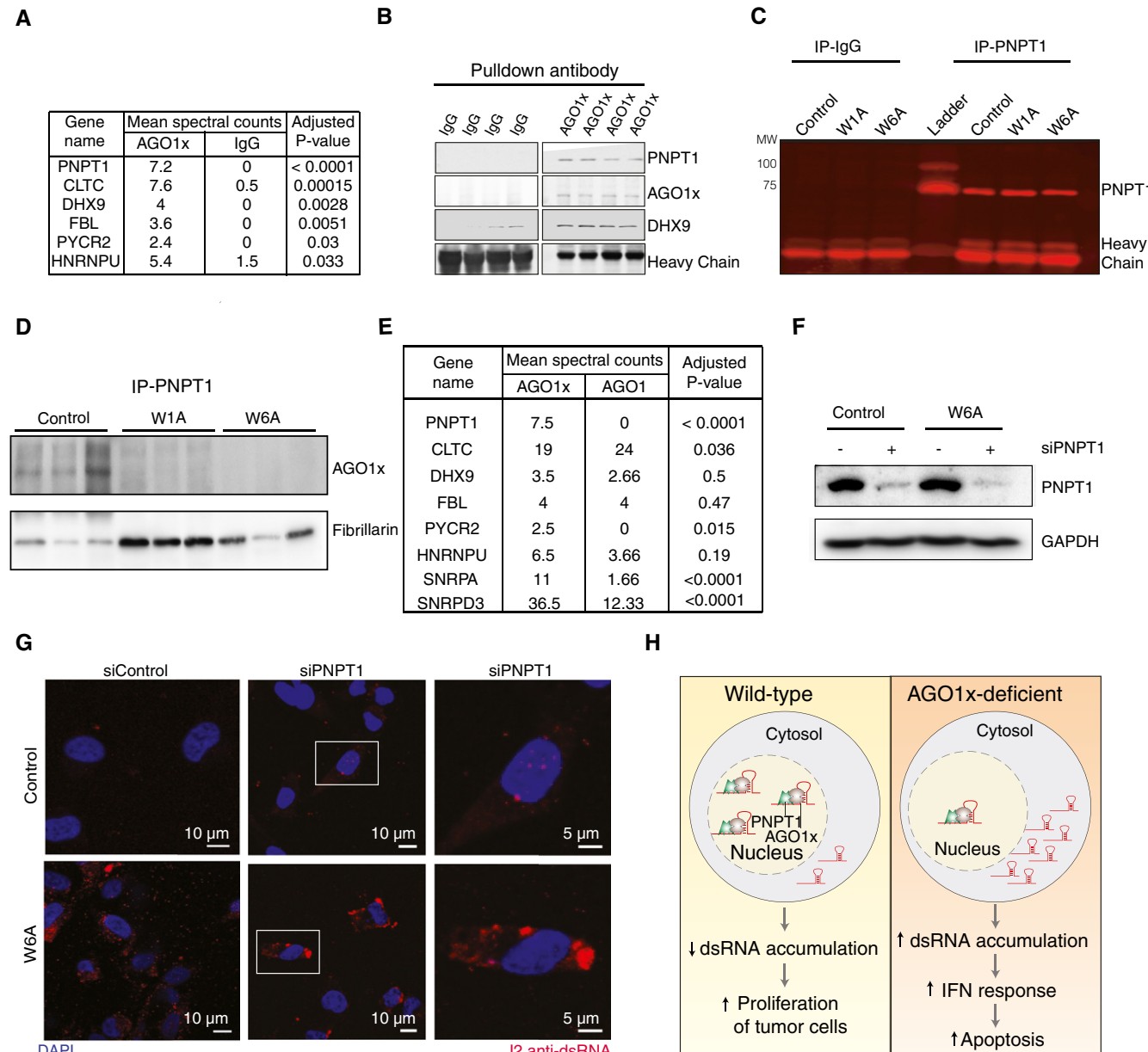

**Figure 6. AGO1x interacts with PNPT1 to buffer endogenous dsRNA levels and regulate growth.**

A   Summary of mass spectrometric analysis of AGO1x ($n = 5$) or IgG ($n = 2$) IP from control cells. *P*-values were calculated using Fisher's exact test and corrected for multiple testing using the Benjamini–Hochberg method.

B   Western blot analysis to validate the interaction of AGO1x with PNPT1 and DHX9 in control MDA-MB-231 cells. AGO1x pulldowns were compared to IgG pulldowns to verify the relative levels of PNPT1 and DHX9 from four independent biological replicates. The immunoglobulin heavy chain band served as a loading control.

C, D   Validation of AGO1x-PNPT1 interaction by reciprocal pulldown. Immunoprecipitation of PNPT1 from control and mutant MDA-MB-231 cells and Western blotting to validate the specificity of PNPT1 antibody in comparison with IgG control (C). Presence of AGO1x and fibrillarin in PNPT1 immunoprecipitates. Three independent biological replicates were used, and the eluted proteins were analyzed by Western blotting (D).

E   *In vitro* interaction assay of isolated nuclear extracts from control MDA-MB-231 cells with FLAG-tagged AGO1 or FLAG-tagged AGO1x immunoprecipitated from cells stably expressing the constructs. Table indicates the peptide counts from a second round of immunoprecipitation with anti-FLAG antibody and subsequent analysis by LC-MS/MS. *P*-values are from Fisher's exact test, corrected for multiple testing using the Benjamini–Hochberg method.

F, G   Western blot analysis to verify the depletion of PNPT1 upon siRNA treatment (F) and a representative immunofluorescence image of siPNPT1/siControl-treated control and W6A mutant MDA-MB-231 cells stained with J2 antibody (red). DAPI was used to mark the nucleus (blue) (G). The right-most panels show a magnification of the cells enclosed by the white boxes in the middle panels.

H   Model of AGO1x function. A complex of AGO1x with PNPT1 interacts with nuclear RNAs to prevent the accumulation of structured dsRNAs, supporting proliferation of breast cancer cells. Depletion of AGO1x leads to deleterious accumulation of these RNAs, which in turn leads to activation of interferon (IFN) response and apoptosis.

Source data are available online for this figure.

in some cells (Appendix Fig S8A and B). Moreover, the interaction of PNPT1 with AGO1x is refractory to RNase A treatment, indicating that they interact directly (Appendix Fig S8C). As PNPT1 was the most specific interactor of AGO1x, we tested whether this protein also affects dsRNA processing. Indeed, the siRNA-mediated depletion of PNPT1 exacerbated the accumulation of dsRNAs in both the parental and the W6A AGO1x mutant MDA-MB-231 cells (Fig 6F and G). These results demonstrate that AGO1x interacts with PNPT1, both proteins repressing the accumulation of endogenous dsRNAs (Fig 6H).

## Discussion

Argonaute proteins form a conserved yet versatile family that has been co-opted in many gene-silencing processes in a variety of organisms (Pratt & MacRae, 2009). RNA immunoprecipitation studies found that the four human Argonaute proteins (AGO1-4) have largely overlapping sets of mRNA targets and thus likely similar functions (Landthaler *et al*, 2008; Hafner *et al*, 2010). The observation that the *AGO1* mRNA is a good substrate for stop codon readthrough (Eswarappa *et al*, 2014) suggested that the functional repertoire of this protein family is still not fully known. The unusually high evolutionary conservation of the *AGO1* 3′ untranslated region (Fig 1A–C), which encodes a C-terminal extension of 33 amino acids in humans, points to an important physiological role. A very recent study (Singh *et al*, 2019) provided experimental evidence for AGO1x presence in mammalian cells. It further showed that, although it does not interact with known effectors of the miRNA-mediated silencing pathways such as the TNRC6B/GW182 proteins, AGO1x still retains the ability to bind mRNA targets, thereby acting as an efficient inhibitor of miRNA-mediated repression. Intrigued by the unusually conserved and unique C-terminal extension of AGO1x, we aimed to further characterize the function of this protein in its physiological context. Analyzing AGO1x expression in a number of cell systems, we confirmed its previously reported presence in HeLa and HEK293 cells, and chose to focus primarily on the breast cancer cell line MDA-MB-231, where we found the highest expression of AGO1x. Although we did not attempt to quantify the relative abundance of the two isoforms, the Western blots in which distinct AGO1 and AGO1x bands are revealed by the AGO1 antibody suggest that even in MDA-MB-231 cells, AGO1x levels are well under the 40% total AGO expression that would be expected based on prior estimates of TR frequency on the *Ago1* transcript (Singh *et al*, 2019). While we cannot explain this discrepancy, which could be due to differences in cellular systems, antibodies, etc., our results are more in line with more general estimates of TR efficiency on various mammalian transcripts, which only reached 7% (Loughran *et al*, 2018). Even for viruses, where TR is integral to the mechanism of virus protein production, the reported TR efficiency is in the range of 5% (Honigman *et al*, 1991). By LC/MS-based analysis of interaction partners of FLAG-tagged AGO1 and AGO1x, we found a strong depletion of TNRC6B/GW182 in the AGO1x compared to the AGO1 pulldown in both MDA-MB-231 and HeLa cells (Appendix Fig S9A). This finding is in agreement with the report of Singh *et al* (Singh *et al*, 2019) and was also confirmed by Western blotting following immunoprecipitation (Appendix Fig S9B). Surprisingly, while endogenous AGO1x

localized to the nucleus, we found the overexpressed FLAG-tagged AGO1x predominantly in the cytoplasm (Fig EV1D–F). A primarily cytoplasmic localization was also apparent in the data of Singh *et al* (Singh *et al*, 2019), even though that study did not specifically evaluate the presence of the AGO1x in different cellular compartments. These observations suggest that AGO1x localization may depend on its overall expression level, an aspect which may be clarified in future work.

To avoid artifacts associated with expression of tagged protein, we focused our study on loss-of-function cell lines, in which we specifically depleted AGO1x by introducing CRISPR/Cas9-mediated mutations in the 3′ UTR, without significantly affecting the expression of canonical AGO1. Using two distinct parental cell lines, and generating from each two distinct derived lines, with two distinct sgRNAs, we found a highly consistent upregulation of dsRNA-mediated interferon response, reduced cell proliferation, and increased apoptosis (Figs 4 and EV3). Expression of cytoplasmic sensors of short dsRNAs was also increased (Fig 5B and C), and image analysis revealed increased numbers of dsRNA foci recognized by the J2 antibody in the mutant lines (Fig 5F and G). RNA sequencing from a J2 antibody pulldown revealed a very heterogeneous population of molecules, consisting of G/C-rich rRNA and mRNA-derived fragments, which have the propensity to form stable secondary structures (Fig 5H–K). We validated the interaction of some of these RNAs with AGO1x by IP followed by qRT–PCR (Fig 5L). Their similar enrichment in the IP of AGO1 (Appendix Fig S10A and B) is consistent with a general propensity of AGO proteins to bind RNAs (Chi *et al*, 2009; Hafner *et al*, 2010; Leung *et al*, 2011; Helwak *et al*, 2013). In contrast, we identified PNPT1 as a direct interaction partner of AGO1x, but not of AGO1 (Fig 6C–E). The siRNA-mediated depletion of PNPT1 enhances the accumulation of dsRNAs (Fig 6F and G). PNPT1 is primarily known as a mitochondrial protein, which participates in the hSUV3-PNPT1 complex that degrades dsRNA substrates (Wang *et al*, 2009). However, functions in cell cycle regulation and malignancy have also been suggested (French *et al*, 2007; Sokhi *et al*, 2014). Our data are in line with these latter studies, showing both that PNPT1 localizes close to nuclei (Appendix Fig S8A and B) and that it co-immunoprecipitates with AGO1x from nuclear extracts independent of RNA interactions (Fig 6A–E and Appendix Fig S8C). While the interplay between AGO1x and PNPT1 in dsRNA target recognition and subsequent degradation will need to be further characterized, our data show that these proteins have a key role in clearing dsRNA species, which are detrimental to the proliferative capacity of cells. The targets appear to be heterogeneous, but they include rRNAs, whose precursor and mature levels are reduced in both HeLa and MDA-MB-231 AGO1x mutant lines (Appendix Fig S11A and B). This reduction is consistent with AGO1x impacting the pre-rRNAs, but whether the rDNA transcription or post-transcriptional processing of pre-rRNA steps remains to be clarified. Also reduced in the mutant lines is the relative 40S/60S ratio (Appendix Fig S11C and D), which indicates that the perturbation of rRNA levels in the mutant lines has functional consequences. Indeed, measurements of homopropargylglycine (HPG) incorporation confirmed that total translation was reduced in mutant compared to control cell lines (Appendix Fig S11E and F). Thus, the perinuclear localization of AGO1x, its interaction with rRNAs, and the observed translation changes in the mutant lines indicate that AGO1x is involved in the formation and

subsequent function of the ribosome, a role that appears to be important for the maintenance of cell viability under conditions of high proliferation (Fig 3G).

A previously published structure of human AGO1 (Elkayam *et al*, 2017) indicates that the C-terminal extension peptide may interfere with the binding to GW182 proteins, which is mediated by binding pockets located in the C-terminally located PIWI domain. Consistently, GW182 proteins were strongly depleted in AGO1x interactome compared to those of AGO1 (Appendix Fig S9). Also partly mediated by the C-terminal carboxylate domain of the protein is the interaction of the 5′ phosphate of the miRNA with AGO1, which could therefore be impaired in AGO1x. However, the results of Singh *et al* do not support this latter hypothesis, as the study reported that AGO1x can still bind miRNAs (Singh *et al*, 2019). Further studies will be needed to clarify whether miRNA-dependent mechanisms contribute to the AGO1x function that we identified in our study.

Although we focused primarily on cancer cell lines, we observed AGO1x expression in non-malignant human tissues as well (Fig EV5). Interestingly, the AGO1x expression in normal tissues appears to be low overall, restricted to a small proportion of cells whose identity remains to be determined. Whether AGO1x plays a role or could be exploited in pathological conditions where the inhibition of the interferon pathway has been associated with poor clinical outcomes (Chiappinelli *et al*, 2015; Gao *et al*, 2016) is unknown. The broad presence of AGO1x in proliferating cancer cells but not in normal human cells may prove important for the specific targeting of this protein as a possible therapeutic strategy.

# Materials and Methods

### Cell culture, transfections, treatments, and common reagents

MDA-MB-231, HEK293FT, HEK293T, and HeLa cells were cultured as described before (Ghosh *et al*, 2015). Other cell lines used for verification of AGO1x levels were cultured as per ATCC guidelines for a single passage prior to cell lysis. All plasmid transfections were performed with Lipofectamine 2000. siRNA transfections were performed with Lipofectamine RNAiMAX (Life Technologies). Western blotting was performed as described earlier (Ghosh *et al*, 2015), and both HRP-labeled and LI-COR IRDye secondary antibodies were used for signal detection. The HRP-labeled secondary antibodies were developed with SuperSignal™ West Pico PLUS Chemiluminescent Substrate (Thermo Fisher Scientific #34580) or with SuperSignal™ West Femto Maximum Sensitivity Substrate (Thermo Fisher Scientific #34095) especially for AGO1x blots in Figs 1D and 3A, and Appendix Fig S4E and for AGO1 blots in Fig 1D and Appendix Fig S4E. For greater resolution of the canonical AGO1 and AGO1x bands, 10% gels were run longer to run out all proteins with MW lower than ~37 kDa. All Western blot images were documented with Azure c600 Gel Documentation System equipped with a 8.3 MP CCD camera. Ruxolitinib (INCB018424) was obtained from Selleckchem (#S1378) and used at a final concentration of 500 nM in DMSO (Sigma #41639). Details of the plasmids used in the study are included in Table 1. The strategy for guide RNA cloning and selection of CRISPR mutants and control lines was followed from a previously published protocol (Ran *et al*, 2013). siRNA and sgRNAs

used in the study are listed in Table 1. Antibodies used for the study are listed in Table 2.

### Genome-wide analysis of C-terminal protein extensions

Putative C-terminal extensions were predicted by taking the mRNA region between the annotated stop codon and the next in-frame stop codon of all RefSeq-annotated (on September 2014) transcripts. PhastCons conservation scores across 45 different vertebrate species for each nucleotide in these regions were downloaded from the UCSC Genome Browser (Kent *et al*, 2002) and averaged across the entire regions of putative translational readthrough. Multiple alignments of the 45 vertebrates genomes were also retrieved from the UCSC Genome Browser, and for Fig 1, the 20 sequences with highest homology to AGO1 extended region were re-aligned using ClustalW (Larkin *et al*, 2007).

### Cell fractionation for localization analysis

Cells were grown in 60-mm dishes for 18–24 h and snap-frozen immediately in liquid nitrogen. Subsequently, the cells were fractionated as described before (Suzuki *et al*, 2010). Fraction lysates obtained were loaded in volume equivalents for each fraction, and Western blots were developed with Pico PLUS Chemiluminescent Substrate except for AGO1x (with Femto Maximum Sensitivity Substrate).

### Immunofluorescence

For immunofluorescence analysis, cells were fixed with 4% paraformaldehyde for 30 min. Subsequently, they were permeabilized and blocked for 30 min with 0.1% Triton X-100 (#T8787, Sigma-Aldrich), 10% goat serum (#16210072, Gibco®, Life Technologies), and 1% BSA (#A9647, Sigma-Aldrich) in PBS (#20012-019, Gibco®, Life Technologies). When AGO1 antibodies were used for staining, the blocking buffer was modified to use donkey serum (D9663-10ML) instead of goat serum. Thereafter, the cells were incubated with primary antibodies (1:100 dilution) in the same buffer at desired dilution overnight at 4°C. Secondary anti-rabbit, anti-goat, or anti-mouse antibodies labeled either with Alexa Fluor® 488 dye (green), Alexa Fluor® 568 dye (orange), Alexa Fluor® 594 dye (red), or Alexa Fluor® 647 dye (far red) fluorochromes (Molecular Probes) were used at 1:500 dilutions. The cells were mounted on a glass slide with VECTASHIELD DAPI (Vector Laboratories), and cells were mostly observed and documented with a ZEISS point scanning confocal LSM 700/LSM 800 inverted microscopes with a PLAN APO 40× (NA = 1.3) and 63× (NA = 1.4) oil-immersion objectives. Laser settings and digital gain settings were set so as to avoid any saturated pixel in the resulting capture. Z-stack images were captured wherever mentioned in text.

### Quantification of AGO1x subcellular localization from IF images

Quantitative analysis of confocal z-stacks was performed in MATLAB r2016a (http://www.mathworks.com) with the Image Processing Toolbox (Gonzalez *et al*, 2004) and ImageJ v1.51n (Schneider *et al*, 2012) (http://imagej.nih.gov/ij/). The processing of every z-stack was executed in three steps. First, we detected the

**Table 1.** Plasmids and oligonucleotides used in this study.

| Name | Sequence Info | Company |
|---|---|---|
| si AGO1 | – | siTOOLs Biotech #26523 |
| sgRNA W6A | AAGAAAGCTTTCCAAGCCCC | Microsynth AG |
| sgRNA W1A | GCAGAACGCTGTTACCTCAC | Microsynth AG |
| pSpCas9n(BB)-2A-Puro | – | Addgene #62988 |
| pSpCas9(BB)-2A-GFP | – | Addgene #48138 |
| pIRES-Neo-FLAG/HA-AGO1 | – | Addgene #6798 |
| pIRES-Neo-FLAG/HA-AGO1x | Modified from Addgene #6798, Additional nucleotides appended to the stop codon in #6798: aggcagaacgctgttacctcactggatagaagaaagctttccaagccccaggagctgt gccacccaaatccagaggaagcaaggaggagggaggtggggtag. The stop codon of #6798 modified to tcc. | – |
| pCDH-AGO1 | AGO1 sequence from pIRES-Neo-FLAG/HA-AGO1 was cloned into the pCDH expression vector | System Biosciences #CD527A-1 |
| pCDH-AGO1x | AGO1x sequence from pIRES-Neo-FLAG/HA-AGO1x was cloned into the pCDH expression vector | – |
| ON-TARGETplus Human PNPT1 (87178) siRNA | – | Horizon Discovery |
| FDFT1 Fwd | GACACGTGGGCGACTTATTG | Microsynth |
| FDFT1 Rev | AGCGAGTCCTGGTCCATCTT | Microsynth |
| SCD Fwd | TCCAGAGGAGGTACTACAAACCT | Microsynth |
| SCD Rev | CCGGGGGCTAATGTTCTTGT | Microsynth |
| EFHD2 Fwd | GATGTTCAAGCAGTATGATGCC | Microsynth |
| EFHD2 Rev | CTTGCGGAAGATCAGGAGGAA | Microsynth |
| SEMA6B Fwd | ACCGTTGTCTTCCTGGGTTC | Microsynth |
| SEMA6B Rev | GCCGATACAGTTCTTCATACACC | Microsynth |
| HAS2 Fwd | GCTGAACAAGATGCATTGTGAG | Microsynth |
| HAS2 Rev | ATAGGCAGCGATGCAAAGGG | Microsynth |
| 5.8 S rRNA Fwd | CTTAGCGGTGGATCACTCGG | Microsynth |
| 5.8 S rRNA Rev | AGTGCGTTCGAAGTGTCGAT | Microsynth |
| 28S rRNA Fwd | AGGTAGCCAAATGCCTCGTC | Microsynth |
| 28S rRNA Rev | TTCACCGTGCCAGACTAGAG | Microsynth |
| 18S rRNA Fwd | TGACTCTAGATAACCTCGGG | Microsynth |
| 18S rRNA Rev | GACTCATTCCAATTACAGGG | Microsynth |
| 45S pre-ribosomal RNA Fwd | GTGAAACCTTCCGACCCCTC | Microsynth |
| 45S pre-ribosomal RNA Rev | TACGAGGTCGATTTGGCGAG | Microsynth |
| NUDT21 Fwd | TGTACATGAGCACCGGCTAC | Microsynth |
| NUDT21 Rev | CCTGACGACCCAGTATCTCTG | Microsynth |
| CPSF6 Fwd | TGAGTCCAAGTCTTATGGTTCTGG | Microsynth |
| CPSF6 Rev | CCTCTTCCTTCAGCTTCTAACGA | Microsynth |
| GAPDH Fwd | AATCCCATCACCATCTTCCA | Microsynth |
| GAPDH Rev | TGGACTCCACGACGTACTCA | Microsynth |

location of every nucleus at each layer of the z-stack. For this, we created the nuclei projection mask by applying the maximum function to all blue channel (DAPI) images from the z-stack and subsequently segmenting the resulting image. The mask was further applied to each blue channel image of the z-stack, thus defining the area where nuclei are located. Then, if the nucleus was present in this area, it was segmented. Second, we detected the location of every nucleolus at each layer of the z-stack. We applied the nuclei projection mask to all red channel (Nucleolin) images, to remove the noise outside of the nuclei. Then, for resulting red channel images, we applied the same procedure as for nuclei detection. For assigning nucleoli to corresponding nuclei, we considered the

**Table 2.  Antibodies used in this study.**

| Name | Company |
|------|---------|
| α-Tubulin | Calbiochem, # CP-06 |
| Nucleolin | Thermo Scientific, #396400 |
| ERP72 | BD Biosciences, #610970 |
| Lsm4 | Sigma, #GW22314F |
| SC35 antibody | Abcam, #ab11826 |
| AGO1x | Lucerna-Chem, #RBP 1510 |
| AGO1 | Novus Biologicals, #NB100-2817 |
| p54 | BD Biosciences, #611278 |
| GAPDH | Sigma, #G9295 |
| hnRNP C1/C2 | Santa Cruz, #SC10037 |
| Ki-67 | Dako, #IR626 |
| J2 | Scicons #10010200 |
| PNPT1 AB (4C11) | Novus #NBP2-43725 |
| DHX9 | Bethyl #A300-855A |
| ANTI-FLAG® M2 | Sigma #F3165-.2MG |
| Actin | Santa Cruz #SC1615 |
| Fibrillarin | Bethyl #A303-891A-M |
| FLAG | Sigma #M8823-1ML |
| PARP cleaved | Cell Signaling #5625 |
| Phospho-eIF2α (Ser 51) | Cell Signaling #3398 |
| PARP | Cell Signaling #9542 |
| RELA | Cell Signaling #8242S |
| Phospho-eIF2α (Ser 52) | Santa Cruz #sc-601670 |
| Phospho-PKR | Abcam #ab81303 |
| GFP | Roche #11814460001 |
| Tubulin | Sigma #T6557 |
| eIF2α | Cell Signaling #9722 |

overlap projection masks of nuclei and nucleoli. Finally, for green (AGO1x) channel images, we collected the intensity of pixels belonging to nucleoli and nucleoplasm, respectively.

**Transcriptome profiling with total RNA-seq**

Total RNA was quality-checked on the Bioanalyzer instrument (Agilent Technologies, Santa Clara, CA, USA) using the RNA 6000 Nano Chip (Agilent, Cat# 5067-1511) and quantified by spectrophotometry using the NanoDrop ND-1000 Instrument (NanoDrop Technologies, Wilmington, DE, USA). 1 μg total RNA was used for library preparation with the TruSeq Stranded mRNA Library Prep Kit High Throughput (Cat# RS-122-2103, Illumina, San Diego, CA, USA). Libraries were quality-checked on the Fragment Analyzer (Advanced Analytical, Ames, IA, USA) using the Standard Sensitivity NGS Fragment Analysis Kit (Cat# DNF-473, Advanced Analytical). The average concentration was 128 ± 12 nM. Samples were pooled to equal molarity. Each pool was quantified by PicoGreen fluorometric measurement to be adjusted to 1.8 pM and used for clustering on the NextSeq 500 instrument (Illumina). Samples were

sequenced using the NextSeq 500 High Output Kit (75 cycles) (Illumina, Cat# FC-404-1005). Primary data analysis was performed with the Illumina RTA version 2.4.11 and base-calling software version bcl2fastq-2.20.0.422.

**Total RNA-seq analysis**

Total RNA-seq reads were subjected to 3′ adapter trimming (5′-TGGAATTCTCGGGTGCCAAGG-3′) and quality control (reads shorter than 20 nucleotides or for which over 10% of the nucleotides had a Phred quality score < 20 were discarded). Filtered reads were then mapped to the human transcriptome based on genome assembly hg19 and transcript annotations from RefSeq with the segemehl software (Hoffmann *et al*, 2009), v0.1.7-411, allowing a minimum mapping accuracy of 90%. Transcript counts were calculated based on uniquely mapped reads and used for differential expression analysis with DESeq2 (Love *et al*, 2014).

For the splicing analysis, filtered reads were mapped to the human genome (hg19) with STAR (Dobin *et al*, 2013), v2.6.0c, using default parameters. Read alignments to the transcriptome were visualized using IGV (Robinson *et al*, 2011), v2.4.16.

**dsRNA pulldown and library preparation**

dsRNAs were enriched using J2 antibody (Scicons # 10010200) from cell lysates as performed earlier (Whipple *et al*, 2015). The library preparation started from 5 ng of RNA based on fluorometric assessment for each sample. No other selection (poly(A)+ or ribo-depletion) was performed to allow unbiased detection of rRNAs as well as mRNAs. Standard fragmentation/priming steps were done as before (for total RNA-seq) to obtain cDNA libraries.

**dsRNA-seq differential expression analysis**

RNA-seq reads obtained were subjected to the standard 3′ adapter trimming (5′-GATCGGAAGAGCACACGTCTGAACTCCAGTCAC-3′) and quality control (reads shorter than 20 nucleotides or for which over 10% of the nucleotides had a PHRED quality score < 20 were discarded). Filtered reads were then mapped to the human transcriptome based on genome assembly hg19 and transcript annotations from RefSeq with the segemehl software (Hoffmann *et al*, 2009), v0.1.7-411, allowing a minimum mapping accuracy of 90%. Unmapped reads were additionally mapped to an "artificial transcriptome" composed of consensus sequences for more than 1,000 different repeat elements present in the human genome (including ancestral shared) as defined by the Genetic Information Research Institute Repbase v23.09 (https://www.girinst.org/repbase/). Transcript counts were calculated based on uniquely mapped reads (for mRNAs, rRNAs, and repetitive elements) and multi-mapped reads (for repetitive elements), and used for differential expression analysis with DESeq2 (Love *et al*, 2014).

**Comparison between dsRNA-seq and total RNA-seq**

To identify RNA transcripts enriched or depleted in the dsRNA pulldown sample, we compare the fold changes in transcript abundances between the dsRNA-seq samples from mutant and control cell lines to those observed for total RNA-seq samples. Since fold

changes computed from the different datasets were well correlated, we defined RNA transcripts as being enriched or depleted in dsRNA structures as those that deviated significantly from the linear regression of dsRNA-seq as a function of total RNA-seq fold changes 4(|standardized residuals| > 2.5 SD). Only RNA transcripts consistently enriched/depleted in the comparisons between the two different mutant cell lines and the control were considered for further analysis.

### Identification of AGO1x by targeted LC-MS

For each sample, $5 \times 10^6$ cells were lysed and AGO1x-affinity-purified using the specific antibody. After washing, the beads and the associated proteins were reduced with 5 mM TCEP for 30 min at 60°C and alkylated with 10 mM chloroacetamide for 30 min at 37°C. Subsequently, the protein sample was digested by incubation with sequencing-grade modified trypsin (1/50, w/w; Promega, Madison, Wisconsin) overnight at 37°C. Finally, peptides were desalted on C18 reversed-phase spin columns according to the manufacturer's instructions (Microspin, Harvard Apparatus), dried under vacuum, and stored at −80°C until further processing. Next, 0.1 µg of peptides of each sample was subjected to targeted MS analysis. Therefore, 6 peptide sequences specific for AGO1 and AGO1x were selected and imported into the Skyline software V2.1, https://brendanx-uw1.gs.washington.edu/labkey/project/home/software/Skyline/begin.view. Then, a mass isolation list comprising the precursor ion masses with charge 2 and 3+ of all peptides was exported and used for parallel reaction monitoring (PRM) (Peterson et al, 2012) and quantification on a Q Exactive HF platform. In brief, peptide separation was carried out using an EASY-nLC 1000 system (Thermo Fisher Scientific) equipped with a RP-HPLC column (75 µm × 30 cm) packed in-house with C18 resin (ReproSil-Pur C18-AQ, 1.9 µm resin; Dr. Maisch GmbH, Ammerbuch-Entringen, Germany) using a linear gradient from 95% solvent A (0.1% formic acid and 2% acetonitrile) and 5% solvent B (98% acetonitrile and 0.1% formic acid) to 45% solvent B over 60 min at a flow rate of 0.2 µl/min. $3 \times 10^6$ ions were accumulated for MS1 and MS2 and scanned at a resolution of 60,000 FWHM (at 200 $m/z$). Fill time was set to 150 ms for both scan types. For MS2, a normalized collision energy of 28% was employed, the ion isolation window was set to 0.4 Th, and the first mass was fixed to 100 Th. All raw files were imported into Skyline for protein/peptide quantification. To control for variation in injected sample amounts, samples were normalized using the total ion current from the MS1 scans. Finally, all generated raw files were subjected to standard database searching to validate the peptide identity. Therefore, the acquired raw files were converted to the Mascot generic file (mgf) format using the msconvert tool (part of ProteoWizard, version 3.0.4624 (2013-6-3)). Using the Mascot algorithm (Matrix Science, version 2.4.0), the mgf files were searched against a decoy database containing normal and reverse sequences of the predicted Swiss-Prot entries of Homo sapiens (www.uniprot.org, release date 29/06/2015), the C-terminal extension in AGO1x, and commonly observed contaminants (in total 41,159 protein sequences) generated using the SequenceReverser tool from the MaxQuant software (version 1.0.13.13). The precursor ion tolerance was set to 10 ppm, and fragment ion tolerance was set to 0.02 Da. The search criteria were set as follows: Full tryptic specificity was required (cleavage after arginine residues

unless followed by proline), 1 missed cleavage was allowed, carbamidomethylation (C) was set as fixed modification, and oxidation (M) was set as variable modifications. Next, the database search results were imported to the Scaffold Q+ software (version 4.3.3, Proteome Software Inc., Portland, OR), and the peptide and protein false identification rate was set to 1% based on the number of decoy hits.

### Global proteome and phosphoproteome analysis by shotgun LC-MS

For each sample, $5 \times 10^6$ cells were washed twice with ice-cold 1× phosphate-buffered saline (PBS) and lysed in 100 µl urea lysis buffer (8 M urea (AppliChem), 0.1 M ammonium bicarbonate (Sigma), and 1× PhosSTOP (Roche)). Samples were vortexed, sonicated at 4°C (Hielscher), shaken for 5 min on a thermomixer (Eppendorf) at room temperature, and centrifuged for 20 min at 4°C in full speed. Supernatants were collected, and protein concentration was measured with BCA Protein Assay Kit (Invitrogen). Per sample, a total of 300 µg of protein mass were reduced with tris(2-carboxyethyl)phosphine (TCEP) at a final concentration of 10 mM at 37°C for 1 h, alkylated with 20 mM chloroacetamide (CAM, Sigma) at 37°C for 30 min, and incubated for 4 h with Lys-C endopeptidase (1:200 w/w). After diluting samples with 0.1 M ammonium bicarbonate to a final urea concentration of 1.6 M, proteins were further digested overnight at 37°C with sequencing-grade modified trypsin (Promega) at a protein-to-enzyme ratio of 50:1. Subsequently, peptides were desalted on a C18 Sep-Pak cartridge (VAC 3 cc, 500 mg, Waters) according to the manufacturer's instructions, split into peptide aliquots of 200 and 25 µg, dried under vacuum, and stored at −80°C until further use.

For proteome profiling, sample aliquots containing 25 µg of dried peptides were subsequently labeled with isobaric tag (TMT 6-plex, Thermo Fisher Scientific) following a recently established protocol (Ahrné et al, 2016). To control for ratio distortion during quantification, a peptide calibration mixture consisting of six digested standard proteins mixed in different amounts was added to each sample before TMT labeling. After pooling the TMT-labeled peptide samples, peptides were again desalted on C18 reversed-phase spin columns according to the manufacturer's instructions (Macrospin, Harvard Apparatus) and dried under vacuum. TMT-labeled peptides were fractionated by high-pH reversed-phase separation using a XBridge Peptide BEH C18 Column (3.5 µm, 130 Å, 1 × 150 mm, Waters) on an Agilent 1260 Infinity HPLC System. Peptides were loaded on column in buffer A (ammonium formate (20 mM, pH 10) in water) and eluted using a two-step linear gradient starting from 2 to 10% in 5 min and then to 50% (v/v) buffer B (90% acetonitrile/ 10% ammonium formate (20 mM, pH 10)) over 55 min at a flow rate of 42 µl/min. Elution of peptides was monitored with a UV detector (215 nm, 254 nm). A total of 36 fractions were collected, pooled into 12 fractions using a post-concatenation strategy as previously described (Wang et al, 2011), dried under vacuum, and subjected to LC-MS/MS analysis.

For phosphoproteome profiling, sample aliquots containing 200 µg of dried peptides were subjected to phosphopeptide enrichment using IMAC cartridges and a BRAVO AssayMAP liquid handling platform (Agilent) as recently described (Post et al, 2017).

The setup of the µRPLC-MS system was as described previously (Ahrné *et al*, 2016). Chromatographic separation of peptides was carried out using an EASY nano-LC 1000 system (Thermo Fisher Scientific), equipped with a heated RP-HPLC column (75 µm × 30 cm) packed in-house with 1.9 µm C18 resin (Reprosil-Pur C18-AQ, Dr. Maisch). Aliquots of 1 µg total peptides were analyzed per LC-MS/MS run using a linear gradient ranging from 95% solvent A (0.15% formic acid and 2% acetonitrile) and 5% solvent B (98% acetonitrile, 2% water, and 0.15% formic acid) to 30% solvent B over 90 min at a flow rate of 200 nl/min. Mass spectrometry analysis was performed on a Q Exactive HF mass spectrometer equipped with a nanoelectrospray ion source (both Thermo Fisher Scientific) and a custom-made column heater set to 60°C. 3E6 ions were collected for MS1 scans for no more than 100 ms and analyzed at a resolution of 120,000 FWHM (at 200 *m/z*). MS2 scans were acquired of the 10 most intense precursor ions at a target setting of 100,000 ions, accumulation time of 50 ms, isolation window of 1.1 Th, and at resolution of 30,000 FWHM (at 200 *m/z*) using a normalized collision energy of 35%. For phosphopeptide-enriched samples, the isolation window was set to 1.4 Th and a normalized collision energy of 28% was applied. Total cycle time was approximately 1–2 s.

For proteome profiling, the raw data files were processed as described above using the Mascot and Scaffold software and TMT reporter ion intensities were extracted. Phosphopeptide-enriched samples were analyzed by label-free quantification. Therefore, the acquired raw files were imported into the Progenesis QI software (v2.0, Nonlinear Dynamics Limited), which was used to extract peptide precursor ion intensities across all samples applying the default parameters.

Quantitative analysis results from label-free and TMT quantification were further processed using the SafeQuant R package v.2.3.2. (https://github.com/eahrne/SafeQuant/) to obtain protein relative abundances. This analysis included global data normalization by equalizing the total peak/reporter areas across all LC-MS runs, summation of peak areas per protein and LC-MS/MS run, followed by calculation of protein abundance ratios. Only isoform-specific peptide ion signals were considered for quantification. The summarized protein expression values were used for statistical testing of differential expression of proteins between conditions. Here, empirical Bayes moderated *t*-tests were applied, as implemented in the R/Bioconductor limma package (http://bioconductor.org/packages/release/bioc/html/limma.html). The resulting per protein and condition comparison *P*-values were adjusted for multiple testing using the Benjamini–Hochberg method.

### Identification of AGO1x-interacting proteins by IP

For identification of the interactors of AGO1x in the cell, MDA-MB-231 cells were lysed in a two-step reaction. In a pre-clearing step, the cells were washed to deplete free cytosolic proteins which would enrich AGO1x in the pulldown while depleting background noise. This was achieved by incubating the cells in a buffer containing 25 mM Tris–HCl, 150 mM KCl, 2 mM EDTA, 0.05% NP40, 1 mM NaF, and 1 mM DTT supplemented with protease inhibitor cocktail for 5 mins and removing the supernatant after centrifugation. Subsequently, the pellets were lysed in a buffer containing 50 mM Tris–HCl pH 7.5, 150 mM NaCl, 5 mM EDTA, 0.5% NP-40, 10% glycerol,

1 mM NaF, and 0.5 mM DTT supplemented with Protease Inhibitor Cocktail—EDTA-Free (Roche). The lysates were clarified by spinning at 2,000 *g* and then incubated overnight with 10 µg of AGO1x or control IgG antibody. Subsequently, 100 µl of Dynabeads® Protein G (Thermo Scientific) was added to each sample for 4 h to facilitate the binding of beads to the antibody. Finally, beads were washed thrice with 50 mM Tris–HCl pH 7.5, 300 mM NaCl, 5 mM MgCl$_2$, 0.05% (v/v) NP-40, and 1 mM NaF and once with PBS. Proteins were eluted with 50 µl of 100 mM glycine pH 2.6, with incubation at room temperature for 5 mins. The eluate was neutralized with 1 M NaOH (1:20) and followed up with either Western blotting or LC-MS analysis. Reciprocal IP for PNPT1 was performed likewise.

### Identification of AGO1x protein interactions by shotgun LC-MS

The setup of the µRPLC-MS system was as described above using an EASY nano-LC 1000 system coupled to a LTQ-Orbitrap Elite (both Thermo Fisher Scientific). Peptide separation was performed as described above. Mass spectrometry analysis was performed on a dual-pressure LTQ-Orbitrap Elite mass spectrometer equipped with a nanoelectrospray ion source and a custom-made column heater set to 60°C. Each MS1 scan (acquired in the Orbitrap) was followed by collision-induced dissociation (CID, acquired in the linear ion trap) of the 20 most abundant precursor ions with dynamic exclusion for 60 s. Total cycle time was approximately 2 s. For MS1, 1E6 ions were accumulated in the Orbitrap cell over a maximum time of 300 ms and scanned at a resolution of 240,000 FWHM (at 400 *m/z*). MS2 scans were acquired at a target setting of 10,000 ions, accumulation time of 25 ms, and rapid scan rate using a normalized collision energy of 35%. The preview mode was activated, and the mass selection window was set to 2 Da.

For data analysis, the acquired raw files were converted to the Mascot generic file (mgf) format using the msconvert tool (part of ProteoWizard, version 3.0.4624 (2013-6-3)). Using the Mascot algorithm (Matrix Science, version 2.4.0), the mgf files were searched against a decoy database containing normal and reverse sequences of the predicted Swiss-Prot entries of Homo sapiens (www.uniprot.org, release date 29/06/2015), the C-terminal extension in AGO1x, and commonly observed contaminants (in total 41,159 protein sequences) generated using the SequenceReverser tool from the MaxQuant software (version 1.0.13.13). The precursor ion tolerance was set to 10 ppm, and fragment ion tolerance was set to 0.02 Da. The search criteria were set as follows: Full tryptic specificity was required (cleavage after arginine residues unless followed by proline), 1 missed cleavage was allowed, carbamidomethylation (C) was set as fixed modification, and oxidation (M) was set as variable modifications. Next, the database search results were imported to the Scaffold Q+ software (version 4.3.3, Proteome Software Inc., Portland, OR), and the peptide and protein false identification rate was set to 1% based on the number of decoy hits. Peptide spectrum counts were used for differential protein analysis.

### *In vitro* assay for tagged AGO1 and AGO1x interactions

FLAG-tagged AGO1 and FLAG-tagged AGO1x constructs (listed in Table 1) were used to stably transfect control MDA-MB-231 cells. Once selected for stable expression, these cells were lysed and

immunoprecipitation was carried out with anti-FLAG M2 beads (Sigma-Aldrich) using buffers and conditions exactly as described in the manufacturer's protocol. After IP, the beads were washed with a washing buffer composed of 50 mM Tris–HCl pH 7.5, 500 mM NaCl, 5 mM MgCl$_2$, 0.05% (v/v) NP-40, and 1 mM NaF and once with PBS. The washed beads were then incubated with nuclear fractions isolated from MDA-MB-231 cells as indicated in the earlier method for detection of AGO1x interactors. To identify interactors of either FLAG-tagged protein, the beads were washed after overnight incubation in a buffer composed of 50 mM Tris–HCl pH 7.5, 300 mM NaCl, 5 mM MgCl$_2$, 0.05% (v/v) NP-40, and 1 mM NaF. Subsequently, the proteins were eluted with 50 µl of 100 mM glycine, pH 2.6, by incubation at room temperature for 5 min. The eluate was neutralized with 1 M NaOH (1:20) and followed up with LC-MS analysis. For the analysis of *ex vivo* interactors of the same tagged constructs in a parallel experiment, no further incubation steps were necessary. The samples were directly split into two parts after IP from total cell lysates with anti-FLAG M2 beads (Sigma-Aldrich), washing, and elution with glycine as done earlier (as per manufacturer's protocol). They were either loaded onto SDS–PAGE gels for detection of candidate proteins of interest or analyzed by LC/MS as described earlier.

### qRT–PCR to estimate the abundance of J2 targets in AGO1x-IP

AGO1x IP was performed exactly as described earlier. Following elution and normalization of pH, Tri Reagent (# 93289-100ML Sigma-Aldrich) was added to the eluate, and subsequently, RNA was purified using standard protocol as described by the manufacturer (# 93289-100ML Sigma-Aldrich). 50 ng of AGO1x-IPed RNA was used for reverse transcription following the manufacturer's protocol and cycling conditions (High-Capacity cDNA Reverse Transcription Kit, Thermo Fisher Scientific). Subsequently, the RT reaction was diluted fourfold with water and subjected to qPCR in a 96-well format, using primers specific to individual genes and GoTaq® qPCR Master Mix (Promega). The incubation and cycling conditions were set as described in the kit, and the plates were analyzed in a StepOnePlus Real-Time PCR System (Thermo Scientific).

### Patient samples

Sixteen consecutive breast cancers tissues and tumor-free tissues from six organs (breast, lung, kidney, prostate, stomach, and colon) were retrieved from the archive of the Institute of Pathology at the University Hospital Basel (Basel). Samples were anonymized prior to analysis, and the approval for the use of these samples has been granted by the local ethics committee (Number: 2016-01748).

### Immunohistochemistry

Immunohistochemical (IHC) staining for AGO1x and Ki-67 was performed on 4-µm sections of FFPE tissue using primary antibodies against anti-AGO1x (Lucerna-Chem; clone RBP 1510, dilution 1:100, citrate buffer pH 6.0 antigen retrieval) and anti-Ki-67 (Dako; clone IR626, dilution 1:200, citrate buffer pH 6.0 antigen retrieval). Staining procedures were performed on Leica Bond III autostainer using Bond ancillary reagents and a Refine Polymer Detection system according to the manufacturer's guidelines. Immunoreactivity for

AGO1x and Ki-67 was performed semi-quantitatively as the number of positive tumor cells over the total number of tumors as previously described (Sepe *et al*, 2016). All slides were evaluated by a trained pathologist (LMT).

### Soft agar colony formation assay

Soft agar assays were carried out in 6-well plates previously coated with a 5 ml layer of medium containing 40% Dulbecco's modified Eagle's medium 2× (Thermo Fisher Scientific), 10% FBS (Gibco, Thermo Fisher Scientific), 10% TPB buffer (Thermo Fisher Scientific), and 0.5% Noble agar (Difco). $5 \times 10^4$ cells were diluted in 1.5 ml of 0.16% agar containing medium, layered onto the bed of 0.5% agar in duplicate, and left growing for 2 weeks in the incubator. Multiple fields were imaged using an inverted microscope, and cell colonies were counted using ImageJ software.

### Proliferation assay

Cell growth was assayed using the xCELLigence system (RTCA, ACEA Biosciences, San Diego, CA). Background impedance of the xCELLigence system was measured for 12 s using 50 µl of room temperature cell culture media in each well of E-plate 16. After reaching 75% confluence, the cells were washed with PBS and detached from the flasks using a short treatment with trypsin/EDTA. 10,000 cells were dispensed into each well of an E-plate 16. Growth and proliferation of the cells were monitored every 15 min up to 48 h via the incorporated sensor electrode arrays of the xCELLigence system, using the RTCA-integrated software according to the manufacturer's parameters.

### Migration assay

Migration assay was performed with CIM plates using the xCELLigence system (RTCA, ACEA Biosciences). $3 \times 10^4$ cells were plated in each well according to manufacturer's instructions, and migration was monitored up to 48 h after seeding using the RTCA-integrated software, according to the manufacturer's protocols (Limame *et al*, 2012).

### Sphere formation assay

$10^3$ cells were resuspended in 4 ml of STEM medium containing DMEM/F12 (Thermo Fisher Scientific), B27 supplement (Thermo Fisher Scientific, 1X), EGF (Sigma-Aldrich, 20 ng/ml), and FGF (Sigma-Aldrich, 10 ng/ml). The cells were then plated in T25 flasks pre-coated with 1% Noble agar (Difco). Fresh aliquots of medium were added every 3 days, and after 10 days, the spheres were observed and counted on an Olympus CKX41 inverted microscope equipped with a SC 30 digital camera (Olympus) and counted using the ImageJ software.

### Apoptosis assay

Steady-state apoptosis assays were performed with the Dead Cell Apoptosis Kit with Annexin V Alexa Fluor™ 488 and propidium iodide (PI) obtained from Thermo Fisher Scientific (#V13241), and assays were performed according to manufacturer's protocol.

Briefly, approx. $10^6$ cells were seeded in a 60-mm cell culture plate and incubated at 37°C in an incubator as normal. Subsequently, the cells were harvested with trypsin–EDTA after overnight growth (between 16 and 18 h) and stained with annexin V conjugated with Alexa 488. For detection of dead/necrotic cells, PI counterstaining was also performed. Negative staining controls and single dye staining controls were also made for each cell type for offline gating analyses with FlowJo®.

All samples were processed in a BD FACSCanto II analyzer.

## Viral stocks and virus infection

Plasmid carrying a green fluorescent protein (GFP)-SINV genomic sequence (a kind gift of C. Saleh) was linearized with XhoI as in Girardi et al (2013). It was used as a substrate for in vitro transcription using mMESSAGE mMACHINE capped RNA transcription kit (Ambion, Thermo Fisher Scientific Inc.) following the manufacturer's instructions. GFP expression is driven by duplication of the subgenomic promoter. SINV-GFP viral stocks were prepared in BHK21 cells, and titers were measured by plaque assay. Cells were infected with SINV-GFP at a MOI of 1 or 10−1, and samples were harvested at 48 h post-infection (hpi).

## Analysis of viral titer by plaque assay

Vero R cells were seeded in 96-well plate format and were infected with 10-fold serial dilution infection supernatants for 1 h. Afterward, the inoculum was removed, and cells were cultured in 2.5% carboxymethyl cellulose for 72 h at 37°C in a humidified atmosphere of 5% $CO_2$. Plaques were counted manually under the microscope, and viral titer was calculated according to the formula: PFU/ml = #plaques/(Dilution×Volume of inoculum).

## Western blot analysis of virus-infected cells

Proteins were extracted from cells and homogenized in 350 μl of lysis buffer (50 mM Tris–HCl pH 7.5, 150 mM NaCl, 5 mM EDTA, 1% Triton X-100, 0.5% SDS, and protease inhibitor "cocktail" (cOmplete Mini; Roche Diagnostics)). Proteins were quantified by the Bradford method, and 15–20 μg of total protein extract was loaded on 4–20% Mini-PROTEAN® TGX™ Precast Gels (Bio-Rad) and transferred into Immobilon-P membrane (Millipore). Membranes were blocked with 5% milk and probed with antibodies. The HRP-labeled secondary antibodies were developed with Super Signal™ West Femto Maximum Sensitivity Substrate (Thermo Fisher Scientific #34095). Western blot images were acquired with Fusion FX7 acquisition system equipped with a DarQ9 CCD camera. Antibodies used for these assays are also appended to Table 2.

## qRT–PCR to estimate the abundance of J2 targets in AGO IP in cytosol/nuclear fractions

For detection of dsRNA targets in AGO1 immunoprecipitates, MDA-MB-231 cells were lysed in a two-step reaction to separate the crude cytosolic fraction from the nuclear fraction. This was achieved by incubating the cells in a buffer containing 25 mM Tris–HCl, 150 mM KCl, 2 mM EDTA, 0.05% NP40, 1 mM NaF,

and 1 mM DTT supplemented with protease inhibitor cocktail for 5 min and collecting the supernatant after centrifugation as the cytoplasmic fraction. Subsequently, the pellets were lysed in a buffer containing 50 mM Tris–HCl pH 7.5, 150 mM NaCl, 5 mM EDTA, 0.5% NP-40, 10% glycerol, 1 mM NaF, and 0.5 mM DTT supplemented with Protease Inhibitor Cocktail—EDTA-Free (Roche) and collected as the nuclear fraction. The lysates were clarified by spinning at 2,000 g and then incubated overnight with 5 μg of AGO1 antibody in each tube. Subsequently, 100 μl of Dynabeads® Protein G (Thermo Scientific) was added to each sample for 4 h to facilitate the binding of beads to the antibody. Finally, beads were washed thrice with 50 mM Tris–HCl pH 7.5, 300 mM NaCl, 5 mM $MgCl_2$, 0.05% (v/v) NP-40, and 1 mM NaF and once with PBS. Following the manufacturer's protocol, TRI Reagent (#93289-100ML Sigma-Aldrich) was directly added to the eluted protein RNA mix (elution with glycine and normalization of pH with NaOH like for other IP reactions) for the purpose of extraction of RNA that was immunoprecipitated in each sample and processed accordingly to purify the RNA from each cellular fraction. 20 ng of IPed RNA from each nuclear/cytosolic fraction was used for reverse transcription following the manufacturer's protocol and cycling conditions (High-Capacity cDNA Reverse Transcription Kit, Thermo Fisher Scientific). Subsequently, the RT reaction was diluted fourfold with water and subjected to qPCR in a 96-well format, using primers specific to individual genes and GoTaq® qPCR Master Mix (Promega). The incubation and cycling conditions were set as described in the kit, and the plates were analyzed in a QuantStudio 3 PCR system (Thermo Scientific).

## Multiplexed caspase assay for detection of caspase 8/9/3 activity in cells

Simultaneously monitoring of key caspases involved in apoptosis, the initiator caspases (caspase 8 and caspase 9) and the executioner caspase 3, was performed with kits obtained from Abcam (ab219915). The kit uses DEVD-ProRed™, IETD-R110, and LEHD-AMC as fluorogenic indicators for caspase 3, caspase 8, and caspase 9 activity, respectively. Upon caspase cleavage, three distinct fluorophores are released: ProRed™ (red fluorescence), R110 (green fluorescence), and AMC (blue fluorescence), which was documented in a Tecan i-control multiplate reader using standard settings.

## qRT-PCR for miRNA let-7a in different cells

Real-time analyses by two-step RT–PCR were performed for quantification of miRNA using Thermo Scientific TaqMan chemistry-based miRNA assay system. It was performed with 25 ng of cellular RNA using specific primers for human let-7a (assay ID 000377). U6 snRNA (assay ID 001973) was used as an endogenous control. One third of the reverse transcription mix was subjected to PCR amplification with TaqMan® Universal PCR Master Mix No AmpErase (Thermo Scientific) and the respective TaqMan® reagents for target miRNA. Samples were analyzed in PCR triplicates from two biological replicates. The comparative $C_t$ method which included normalization, by the U6 snRNA, was used for each cell type for plotting of mean values with SD.

## qRT–PCR for estimation of rRNA levels in cells

To estimate the relative levels of the precursor and mature rRNA sequences, we used a two-step qRT–PCR. 100 ng of total RNA extracted from the cells was used to reverse-transcribe to cDNA using the manufacturer's protocol and cycling conditions (High-Capacity cDNA Reverse Transcription Kit, Thermo Fisher Scientific). Subsequently, the RT reaction was diluted fourfold with water and subjected to qPCR in a 96-well format, using primers specific to individual genes and GoTaq® qPCR Master Mix (Promega). The incubation and cycling conditions were set as described in the kit, and the plates were analyzed in a StepOnePlus Real-Time PCR System (Thermo Scientific). Primers used for the PCR are listed in Table 1.

## Quantification of nascent protein synthesis by HPG assay

To quantify nascent protein synthesis as a measure of global translation, we used the non-radioactive metabolic labeling assay Click-iT HPG Alexa Fluor 594 Protein Synthesis Assay Kit (Thermo Fisher Scientific, Cat #C10429). The method is based on the incorporation of L-HPG, an amino acid analog of methionine containing an alkyne moiety, and Alexa Fluor 594 azide. The seeding of MDA-MB-231 cells and incubation of HPG in the medium were performed exactly as done earlier in the laboratory (Guimaraes *et al*, 2020). HeLa cells, in a slight modification from MDA-MB-231 cells, were incubated with HPG for a 30-min duration as per manufacturer's protocol. The signal intensity of the incorporated HPG Alexa Fluor 594 was measured by flow cytometry on a BD LSRFortessa (Beckman Coulter) using the 561 laser, BP filter 610/20 laser. Mean fluorescence intensities were computed from a minimum of 10,000 cells for each sample. The experiment was performed with two biological replicates for HeLa and three biological replicates of MDA-MB-231 cells. In the final labeling steps, quantities of Alexa Fluor azide used for MDA-MB-231 cell experiments were exactly as mentioned in the manufacturer's protocol. For HeLa cells, a slightly modified half of the stipulated amount of the Alexa Fluor azide molecule was utilized.

## Determination of 40S, 60S ribosome abundance in cells

The profile of 40S and 60S ribosomal subunits was obtained according to the protocol of polysome profiling described in the previous section "Polysome analysis of cells" with the following modifications. 40 mM EDTA was added to the polysome lysis buffer and gradient buffer to help dissociate the 40S and 60S ribosome subunits as described in Fleischer *et al* (2006). Further, lysate equivalent to optical density $A_{260} = 3$ was loaded onto the gradient. Finally, the profile was obtained by analyzing the gradient on Piston Gradient Fractionator (BioComp) with a setting of piston speed of 0.3 mm/s. The area under curve for 40S and 60S subunits was calculated using ImageJ 1.52a software.

## Analyzing AGO1x-PNPT1 interaction in the presence of RNase A

For identification of RNA dependence of interaction of AGO1x with PNPT1 in the cell, MDA-MB-231 cells were lysed in a two-step reaction as done earlier with slight variation in the KCl/NaCl concentrations. In a pre-clearing step, the cells were washed to deplete free cytosolic proteins which would enrich AGO1x in the pulldown while depleting background noise. This was achieved by incubating the cells in a buffer containing 25 mM Tris–HCl, 100 mM KCl, 2 mM EDTA, 0.05% NP40, 1 mM NaF, and 1 mM DTT supplemented with protease inhibitor cocktail for 5 min and removing the supernatant after centrifugation. Subsequently, the pellets were lysed in a buffer containing 50 mM Tris–HCl pH 7.5, 100 mM NaCl, 5 mM EDTA, 0.5% NP-40, 10% glycerol, 1 mM NaF, and 0.5 mM DTT supplemented with Protease Inhibitor Cocktail—EDTA-Free (Roche) with or without 20 μl RNase A (#EN0531 Thermo Fisher Scientific). The lysates were then incubated for 30 min at 37°C before subsequent overnight incubation with 5 μg of PNPT1 antibody at 4°C. Subsequently, 100 μl of Dynabeads® Protein G (Thermo Scientific) was added to each sample for 4 h to facilitate the binding of beads to the antibody. Finally, beads were washed thrice with 50 mM Tris–HCl pH 7.5, 300 mM NaCl, 5 mM MgCl₂, 0.05% (v/v) NP-40, and 1 mM NaF and once with PBS. Proteins were directly eluted in 30 μl of Laemmli buffer supplemented with DTT (1×) with incubation at 90 for 5 min and followed up with Western blotting.

## Motif activity analysis

The web server ISMARA (https://ismara.unibas.ch/) was used to estimate the activities of different transcription factor binding motifs in the different mRNA-seq samples.

## Gene set enrichment analysis

The tool GSEA v2.2.3 (http://software.broadinstitute.org/gsea/index.jsp) was used to calculate the enrichment of gene sets derived from the KEGG pathway database and the Hallmark collection. To estimate the significance of the enrichment, the number of permutations was set to 1,000 and the permutation type was set to gene sets.

## Quantification of blotted band intensity with ImageJ

Gels were converted to grayscale and quantified with ImageJ version 1.53a. Briefly, the rectangle tool from ImageJ was used to draw a frame around the largest band of a row, which was measured as per software manual. All band intensities were then subtracted from 255. For each gel, regions outside the bands were also quantified for background estimation, and the corresponding bands were adjusted with respect to background. Unless mentioned otherwise, all band intensities were normalized to the band intensities of loading controls such as GAPDH. Finally, all samples were compared to the control cell line, where the intensity was taken as unit value.

## Statistical analysis

Statistical analysis was performed with Prism 7.0c (GraphPad). For the different statistical tests performed, a $P < 0.05$ was considered significant.

# Data availability

RNA sequencing data from this study have been submitted to the Sequence Read Archive under the accession number PRJNA447929. The data are accessible in the BioProject portal of NCBI at https://

www.ncbi.nlm.nih.gov/bioproject/PRJNA447929. The source code to replicate genomics analysis presented in this study is available from Zenodo at https://zenodo.org/record/3711079. The MS proteomics data have been deposited to ProteomeXchange with the identifier PXD009401. The data are accessible at http://proteocen tral.proteomexchange.org/cgi/GetDataset?ID = PXD009401.

**Expanded View** for this article is available online.

## Acknowledgements

We thank Alexandra Gnann for support with cloning, Dr. Noam Stern-Ginossar for suggestions of the interferon pathway experiments, Christina J. Herrmann for help with the title graphics, and members of the Zavolan group for comments and suggestions. We would also like to thank Philippe Demougin from the Genomics Facility Basel for genomic library preparation, and the sciCORE team for their maintenance of the HPC facility at the University of Basel. This work was funded by the Swiss National Science Foundation NCCR project "RNA & Disease" (grant 51NF40_141735), by a Research Grant from the University of Basel to S.G., and by a SystemsX.ch Transitional Postdoctoral Fellowship (grant 51FSP0_157344) and a Novartis University of Basel Excellence Scholarship for Life Sciences to J.C.G. Additional funding sources include Swiss National Science Foundation (Ambizione grant number PZ00P3_168165) to S.P. and Swiss Cancer League (KFS-3995-08-2016 and KLS-3639-02-2015) to S.P. and L.M.T. G.M. acknowledges funding by the Deutsche Forschungsge-meinschaft (DFG, SFB960). S. Pf. acknowledges funding by the European Research Council (ERC-CoG-647455 RegulRNA) and the French National Research Agency (ANR-10-LABX-0036_NETRNA).

## Author contributions

JCG, SG, and MZ conceived and supervised the project. SG designed and performed most experiments. JCG performed all computational analysis. SG and JCG analyzed data and interpreted results. ML performed immunohisto-chemistry and helped with oncogenic-related assays. AS performed proteomics analysis. APS generated CRISPR/CAS9 cell lines. AB performed image quan-tification analysis. BD and ShG performed Western blots and characterization of mutants. NM carried out the analysis of ribosomal subunits. ALC provided resources and generated overexpression cell lines. JD set up the immunopre-cipitation protocol and provided resources. GM provided resources and input on the analysis of AGO1x-specific antibody. TM performed viral infection assays. SPf and TM interpreted viral infection assay, and SPf provided resources. SPi and LMT interpreted oncogenic assays and immunohistochem-istry stains and provided resources. SG, JCG, and MZ wrote the manuscript.

## Conflict of interest

The authors declare that they have no conflict of interest.

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
