## [Review Process File · The EMBO Journal]

Prevention of dsRNA-induced interferon signaling by AGO1x is linked to breast cancer cell proliferation

Souvik Ghosh, Joao Guimaraes, Manuela Lanzafame, Alexander Schmidt, Afzal Pasha Syed, Beatrice Dimitriades, Anastasiya Boersch, Shreemoyee Ghosh, Nitish Mittal, Thomas Montavon, Ana Luisa Correia, Johannes Danner, Gunter Meister, Luigi Terracciano, Sebastien Pfeffer, Salvatore Piscuoglio, and Mihaela Zavolan

DOI: [10.15252/embj.2019103922](https://doi.org/10.15252/embj.2019103922)

Corresponding author(s): *Mihaela Zavolan (mihaela.zavolan@unibas.ch)*, *Joao Guimaraes (joao.guimaraes@unibas.ch)*

Review Timeline:

Submission Date:	6th Nov 19
Editorial Decision:	21st Nov 19
Revision Received:	17th Mar 20
Editorial Decision:	14th Apr 20
Revision Received:	8th Jun 20
Editorial Decision:	23rd Jun 20
Revision Received:	27th Jun 20
Accepted:	7th Jul 20

Editor: Stefanie Boehm

Transaction Report:

Prof. Mihaela Zavolan
University of Basel
Biozentrum
Klingelbergstrasse 50/70
Basel, Basel-Stadt CH-4056
Switzerland

21st Nov 2019

Re: EMBOJ-2019-103922

AGO1x prevents dsRNA-induced interferon signaling to promote proliferation of breast cancer cells

Dear Prof. Zavolan,

Thank you for submitting your study on the role of AGO1x in dsRNA-induced interferon signaling for consideration by The EMBO Journal. We have now received three referee reports on your study, which are included below for your information.

As you will see, the reviewers express an overall interest in the study, but they also raise several major concerns and technical issues. In particular, referee #2 and #3 find that additional experimental proof is required to support the proposed link between dsRNA accumulation and the observed phenotypes, as well as further clarifying the role of rRNA hereby. In addition, all referees note that a direct comparison of AGO1x to AGO1 interactors is needed to support the proposed AGO1x-specificity for certain candidates, in particular for PNPT1. Furthermore, the differences to the recent study on AGO1x by Singh et al. needs to be addressed in more detail and some experimental comparison of the two systems provided, in particular to address the divergent sub-cellular localization and cell line-specific expression of AGO1x.

Should you be able to address the key points raised by the referees, as well as the numerous more specific concerns, then we would like to invite you to prepare and submit a revised manuscript. Please note that EMBO Journal policy allows only a single round of major revision. We are aware that addressing all issues raised by the referees will likely require a substantial amount of experimental work and include experiments with a potentially unknown outcome. We can extend the revision time up to a total of six months in certain cases or offer to discuss a transfer of the study to another EMBO Press journal, but it is nonetheless important to clarify all key concerns at this stage. Please feel free to contact me should you have any further questions regarding the revision.

We generally allow three months as standard revision time. As a matter of policy, competing manuscripts published during this period will not negatively impact on our assessment of the conceptual advance presented by your study. However, we request that you contact the editor as soon as possible upon publication of any related work, to discuss how to proceed.

Thank you for the opportunity to consider your work for publication. I look forward to your revision.

Kind regards,

Stefanie Boehm

Stefanie Boehm
Editor
The EMBO Journal

Referee #1:

Translational readthrough (TR) can generate C-terminally extended proteins. TR has the potential to generate a novel protein isoform with distinct functional properties. TR was initially characterized for translation of viral transcripts but has recently also been observed in human cells (Eswarappa et al., Cell 2014, PMID: 24949972). TR of human Argonate-1 (AGO1) was observed to generate a C-terminally extended isoform of AGO1, termed AGO1x, with dominant negative properties that functions as an inhibitor of microRNA (miRNA) silencing (Singh et al., and Eswarappa, EMBO J 2019, PMID: 31330067). AGO1 is one of the four human Argonate proteins that associates with miRNAs to silence the expression of target RNAs (reviewed by David Bartel Cell 2018, PMID: 29570994). Here, Ghosh, Guimaraes and colleagues suggest a novel function of AGOx to prevent dsRNA-induced interferon signaling in a human cancer cell line. The experiments largely contradict previously published data on the distribution and function of AGOx (Singh et al., and Eswarappa, EMBO J 2019, PMID: 31330067). The authors should comment on these discrepancies and investigate if the observed differences can be attributed to reagents or the use of a different cell line. Most experiments presented in this manuscript are based on insufficiently characterized reagents: a novel AGO1x antibody and 'AGO1x CRISPR cell lines. Experiments are ill-designed and lack essential controls, which could easily result in unfortunate misinterpretations.

The authors have to thoroughly revise their experiments and add appropriate controls to test the stated hypotheses. I do not recommend this manuscript for publication without major revision.

Major comments:

In Figure 1. The authors show expression of AGO1x in two human cancer cell lines (MDA-MB-231 and HeLa) but not in HEK293 cells using a novel antibody generated to specifically recognize the read-through product. The lack of AGO1x in Hek293 compared to Hela cells is surprising, because similar amounts of AGO1x have been detected in both cell lines previously (Singh et al., and Eswarappa, EMBO J 2019, PMID: 31330067). Furthermore, the authors observe localization of AGO1x to the nucleus in contrast to the previously reported cytoplasmic localization. The authors should comment on these discrepancies and use knock-down or knock-out of AGO1 to assess the specificity of their immunostaining.

Figure 2. The authors perform immunofluorescence analysis using their new AGO1x antibody in MDA-MB-231 cells and detect a speckled nuclear staining with some co-localization with Nucleolin. To assess specificity of this signal, the authors could use tools presented in later stages of the manuscript, i.e. 'loss of AGO1x' cells. Because AGO1 is not an essential gene, a straight-forward KO cell line would also be a great control. The authors should also perform staining using an established AGO1 antibody that will enable detection of both AGO1 and AGO1x.

Figure 3. Here the authors report the use of CRISPR/Cas9 genome editing to eliminate the AGO1 read-through product (AGO1x). In panel A the authors forgot to indicate the generated mutations

(from their diagnostic genotyping). The authors should also add a methods section about the detailed characterization of these cell lines. The accompanying western blot seems confusing as the higher band corresponding to AGO1x is clearly visible in all three lanes using the pan_AGO1 antibody, but a signal is absent in the two clonal lines using the AGO1x antibody. The authors should use quantitative imaging technologies like Odyssey or other methods for this analysis. Cell growth assays should be performed with clonal control cell lines that arose from the same initial CRISPR/Cas9 transfection. Mutations by repair mistakes of a guided Cas9 cut generally result in cell clones with variable mutations and deletions that will have different impact on AGO1 read-through translation. The authors should characterize a set of cell lines that include functional AGO1 null, AGO1x null, AGO1 null and no effect on AGO1X expression mutations. This is particularly important as exact homozygous editing events are rare and MDA-MB-231 are in addition aneuploid (ATCC). Alternatively, the authors could use precise CRISPR editing (with designed donor constructs) to eliminate either only the canonical AGO1 STOP codon or only the AGO1x STOP codon. Multiple clonal cell lines have to be tested for both interference and controls. FLAG-tagged Rescue constructs expressing either FLAG-AGO1 or FLAG-AGO1x (as used in Fig. S8) could also be used to address the specificity of the observed effect.

Fig 3g: the authors should use fluorescent secondary antibodies and perform a control staining for normalization.

Figure 4 and 5 a-f. Here, the authors suggest that depletion of AGO1x results in an increased interferon response. Like for experiments presented in Figure 3, this set of experiments requires well established cell lines and control lines. Without thorough characterization of the used cell lines and appropriate controls these experiments are meaningless.

For pull-down experiments, the authors should compare AGO1 pull downs with AGO1x pull downs to identify proteins and RNAs that specifically interact with AGO1x but not AGO1. Using IgG as a control does not allow discrimination of factors associated with AGO1 or AGO1x. Because the authors aim to claim a specific function for AGO1x and not a novel function for overall AGO1, a thorough differential analysis of both isoforms is essential.

Fig. 5 I and j: Why is DHX9 visible as unspecific background in 2 out of 4 lanes of negative IgG pull downs (Fig. j) but not in the mass spec analysis (i)?

Why would PNPT1 knock down increase the abundance of dsRNA in both the parental and the presumable AGO1x mutant cell lines? Is this effect independent of AGO1x?

Minor points:

The authors should revise their introduction to better represent relevant miRNA literature: Argonaute proteins predominantly localize to the cytoplasm and target mRNAs for degradation. Nuclear localization and potential function of human Argonaute proteins has long been a controversial topic in the small RNA field and is not a universally observed phenomenon (reviewed by David Bartel Cell 2018, PMID: 29570994).

Referee #2:

Ghosh and co-workers report evidence for the existence of AGO1x, an unexpected Argonaute isoform, which is proposed to promote proliferation of breast cancer cells by preventing dsRNA-induced interferon signaling. Key findings include:

- evidence for expression of AGO1x in human cell lines and breast cancer samples
- preferential nuclear localization of AGO1x
- AGO1x promotes cell proliferation
- AGO1x depletion increases dsRNA accumulation, interferon response, and apoptosis
- AGO1x interacts with PNPT1

This manuscript comes on the heels of Singh et al, 2019, which also demonstrated the existence of AGO1x and also characterized its biogenesis and functional properties. Together, the two studies provide compelling evidence for the existence of AGO1x, and complementary biological insights that will have a substantial impact on the field.

The manuscript may be improved by addressing the following:

Major Concerns

Conclusions drawn from data in Figs. 1-4 are exciting and compelling. In contrast, the evidence supporting the final model (as illustrated in Fig. 5i) has several distinct weaknesses:

- 1) AGO1x and PNPT1 are suggested to physically interact (Figs. 5i-j and S7a-b) and collaborate to prevent accumulation of 'dsRNA' (Fig. 5l), and yet AGO1x is shown to be nuclear (Fig. 2) while PNPT1 is cytoplasmic (Fig. S7c). How do the authors reconcile this? Is the AGO1x/PNPT1 interaction direct or at least insensitive to RNase A?
- 2) dsRNA staining in Figs. 5d and 5k is shown in only a few cells. Considering these observations are used as a major piece of evidence for cellular dsRNA up-regulation upon Ago1x knockout, more rigorous analysis is suggested. i.e. what is the distribution of dsRNA staining levels when surveying >100 cells for each sample?
- 3) Purified Argonaute proteins bind RNA non-specifically. Is dsRNA binding (Fig. 5h) specific to AGO1x? An AGO1 IP could be included as a control to Fig. 5h since AGO1 levels are >10-fold AGO1x (Fig. 1d) the AGO1 IP should be dominated by RNAs associated with AGO1 (and not AGO1x).
- 4) Evidence that dsRNA accumulation is responsible for the observed changes in IFN response, apoptosis, and cell proliferation upon AGO1x knockout is tenuous. Thus, there remains a substantial possibility that a different mechanism (including altered miRNA function, activation of the RP-MDM2-p53 pathway, or some other unknown pathway) is the true cause of these effects. In the absence of direct evidence linking the observed 'dsRNA' to the observed cellular responses, I strongly suggest downplaying this conclusion.
- 5) Related to the comment above, Fig. 5e shows increased rRNA levels in W1A and W5A cells. Are ribosomal protein or mature ribosome levels also altered in these cells? Considering the nuclear enrichment of AGO1x in Fig. 2b it seems reasonable to suggest a role in ribosome biogenesis, which if perturbed can initiate nucleolar stress leading to anti-tumor responses.

Minor concerns/suggestions

6) The authors state: "Important for its characterization was the realization that in contrast to the ectopically-expressed AGO1x, which has cytoplasmic localization both in our and in the study of Singh et al. (Singh et al., 2019), endogenous AGO1x is found primarily in the nucleus." This statement does not discuss or acknowledge results in Fig. S4 of Singh et al., 2019, which appear to show cytoplasmic distribution of endogenous AGO1x.

7) Similarly, Singh et al., 2019 report that AGO1x makes up 40% of the endogenous AGO1 in HEK293 cells, and appears to be similarly expressed in HeLa cells. These results stand in contrast to the Western blot in Fig. 1d. This difference should be mentioned in some way.

8) Where is PNPT1 in Fig. S8a?

9) Why is the expression change of IFIH1 not shown in Fig. 5b?

10) Describing rRNA as 'dsRNA' is confusing/inaccurate because, although it does contain a substantial amount of base-paired helical structures, rRNA is not double stranded.

11) The authors state: "PNPT1, whose siRNA-mediated depletion replicates the AGO1x depletion phenotype and also leads to the accumulation of dsRNAs." This statement seems incongruent with data in Fig. 5k, which appears to show more dsRNA staining in the W6A sample than either siPNPT1 sample.

12) Fig. S2a: the AGO1x band appears to run much slower than the AGO1 band (in comparison with Fig. S2b). How did the authors verify that this band corresponds to AGO1x?

13) It would be reasonable to suggest that the TR extension reduces miRNA affinity (Fig. S8c) by moving the C-terminal carboxyl group, which interacts with the miRNA 5' phosphate via a lysine and ordered water molecule in crystal structures of canonical human Ago proteins.

Referee #3:

In this manuscript Ghosh et al. report that a highly conserved TR isoform of AGO1, AGO1x, localizes exclusively to the nucleus, in the vicinity of nucleoli, where it prevents accumulation of double stranded RNAs by an unknown mechanism and thus, supposedly, mitigates the interferon response and apoptosis. A similar role has also been proposed for Polyribonucleotide Nucleotidyltransferase 1, a newly identified, specific binding partner of AGO1x. Finally, they also showed that AGO1x is overrepresented in highly proliferative breast cancer cells and speculate that targeting AGO1x could potentially represent a new direction for limiting tumor growth.

This is a clearly written, technically well-executed, fairly interesting story supported by numerous experiments. However, it suffers from 1) insufficient detail (I've had an impression that it was originally written for some "short-format" journal) forcing the reader to spend quite some time trying to understand what this or that result means and/or how exactly this or that experiment was executed, and 2) the lack of any mechanical insight into the presented observations. Below I summarize all my concerns (in the order of their appearance in the text) that, in my opinion, should be addressed to improve the quality of this study before its prospective acceptance.

1) I suggest the authors to go through the entire manuscript, a result after result, a figure after

figure (especially Sup. Figures but not only those), and explain in greater detail how each experiment was done and what can be seen in each figure (expected vs. unexpected and why; many proteins are listed with no description what they do in cells and why they are expected to change their expression in this or the other direction, etc.) with a non-expert reader in mind. It will improve the clarity and readability of your work by a great margin. Also, label the pages and lines, please.

2) Fig. S5a is placed out of order in the main text; in between S2 and S3.

3) "Endogenous AGO1x localizes" Fig. 1b should be 2b.

4) Fig.3a; the AGO1x strip is smudgy compared to others shown in the work. Since it is a fairly important result I suggest repeating it aiming for a nicer western blot image.

5) I am not convinced that the loss of AGO1x results in increased apoptosis. The differences in Fig. 4e look marginal to me; protein levels of caspases (actually only CASP8 is shown in Fig. 4d) do not dramatically increase either. Other experiments, like cleavage of PARP-1, should be performed to prove or disprove this point.

6) Figures 4c,d and S6e; please explain how the caspases should behave in your opinion and why. Why does CASP9 differ between MDA and HeLa lines?

7) "Hypothesizing that the activation of the interferon response observed upon AGO1x inhibition is due to the accumulation of double-stranded RNAs (dsRNAs) (Field et al , 1967; Lampson et al , 1967), we next examined the levels of intracellular dsRNA sensors (Schneider et al , 2014). We found that their expression was indeed increased, both at the mRNA and at the protein levels (Fig. 5a, b). In addition, using a previously published method (Post et al , 2017), we uncovered evidence for protein phosphorylation events downstream of interferon signaling, in EIF2AK2 and ADAR (Fig. 5c)." This is a typical example of the "go figure yourself" approach mentioned above. What are the sensors (how they operate in a few words) and what are not in the corresp. figure?, why the protein levels of IFIH1 are not shown in 5b?, what "method"?, phosphorylation of EIF2AK2 and ADAR means what?....

8) Figure 5d, k - signal quantifications would help.

9) The fact that the source of dsRNA lies predominantly, if not solely (based on Fig. 5e), in rRNA is rather unexpected. The authors propose that: "...the increased demand for ribosome biogenesis in rapidly dividing cells requires the readthrough AGO1x isoform for resolving dsRNA structures in rRNAs and other molecules."; also rather unexpected, but why not. Since no other explanation has been offered, either for the AGO1x role in dsRNA metabolism in the nucleus, or for its contact with PNPT1, I suggest the authors to explore this idea in greater detail. If correct, ribosome biogenesis should be affected in AGO1x null (as well as PNPT1 knocked-down) cells, which can be easily tested with a bunch of well-established approaches.

10) What could be reason for the observed differences between endogenously vs. ectopically expressed AGO1x? Any idea?

Referee #1:

Translational readthrough (TR) can generate C-terminally extended proteins. TR has the
potential to generate a novel protein isoform with distinct functional properties. TR was initially
characterized for translation of viral transcripts but has recently also been observed in human
cells (Eswarappa et al., Cell 2014, PMID: 24949972). TR of human Argonaute-1 (AGO1) was
observed to generate a C-terminally extended isoform of AGO1, termed AGO1x, with dominant
negative properties that functions as an inhibitor of microRNA (miRNA) silencing (Singh et al.,
and Eswarappa, EMBO J 2019, PMID: 31330067). AGO1 is one of the four human Argonaute
proteins that associates with miRNAs to silence the expression of target RNAs (reviewed by
David Bartel Cell 2018, PMID: 29570994). Here, Ghosh, Guimaraes and colleagues suggest a
novel function of AGOx to prevent dsRNA-induced interferon signaling in a human cancer cell
line. The experiments largely contradict previously published data on the distribution and
function of AGOx (Singh et al., and Eswarappa, EMBO J 2019, PMID: 31330067). The authors
should comment on these discrepancies and investigate if the observed differences can be
attributed to reagents or the use of a different cell line. Most experiments presented in this
manuscript are based on insufficiently characterized reagents: a novel AGO1x antibody and
'AGO1x CRISPR cell lines. Experiments are ill-designed and lack essential controls, which
could easily result in unfortunate misinterpretations.

The authors have to thoroughly revise their experiments and add appropriate controls to test the
stated hypotheses. I do not recommend this manuscript for publication without major revision.

Major comments:

In Figure 1. The authors show expression of AGO1x in two human cancer cell lines (MDA-MB-
231 and HeLa) but not in HEK293 cells using a novel antibody generated to specifically
recognize the read-through product. The lack of AGO1x in Hek293 compared to Hela cells is
surprising, because similar amounts of AGO1x have been detected in both cell lines previously
(Singh et al., and Eswarappa, EMBO J 2019, PMID: 31330067). Furthermore, the authors
observe localization of AGO1x to the nucleus in contrast to the previously reported cytoplasmic
localization. The authors should comment on these discrepancies and use knock-down or
knock-out of AGO1 to assess the specificity of their immunostaining.

The reviewer seems to not have seen our supplementary material which indeed contained
results of the suggested experiments. For instance, in Fig. S2 (revised Fig EV1) we have
addressed the specificity of our antibody by AGO1 knockdown with an siRNA pool as well as by
overexpression of FLAG-tagged AGO1/AGO1x. The signal provided by the AGO1/AGO1x
antibodies are as expected.

Regarding the discrepancies with the study of Singh et al., we think that these have multiple
causes. First, the HEK293 lines were different, HEK293T used by Singh et al. and HEK293FT

used by us. We have assessed AGO1x expression by WB in a number of cells before, and we
have now included a few more lines, in particular the HEK293T and HEK293FT (old Fig. 1d and
S2a, new Fig 1D, EV1, Appendix Fig S2A-B). Although we did observe some bands in the
region expected for AGO1x in HEK293T cells, the most intense band migrated at a higher MW
than expected (further confirmed by AGO1x overexpression) relative to the other cells lines.
Furthermore, in the study of Singh et al. the number of bands and relative intensity in HeLa and
HEK293 cells varied between experiments (compare Figs S4, 2b, c from that study).
Furthermore, in HEK293 cells the immunofluorescence with the AGO1 antibody (Fig. S4) did not
recapitulate the WB shown in Fig. 1e, where AGO1 and AGO1x had comparable intensities. In
addition, some degree of AGO1x nuclear localization can be inferred from Fig. S4 of Singh et al.
as well. Finally, assuming the mechanism proposed by Singh et al., in which the let-7 miRNA
guides the TR of AGO1, we note that the level of the let-7a miRNA is, in fact, lowest in our
HEK293T cells, followed by HeLa and MDA-MB-231 (Appendix Fig S2C). This is also consistent
with our estimates of AGO1x expression in these cells.

Figure 2. The authors perform immunofluorescence analysis using their new AGO1x antibody in
MDA-MB-231 cells and detect a speckled nuclear staining with some co-localization with
Nucleolin. To assess specificity of this signal, the authors could use tools presented in later
stages of the manuscript, i.e. 'loss of AGO1x' cells. Because AGO1 is not an essential gene, a
straight-forward KO cell line would also be a great control. The authors should also perform
staining using an established AGO1 antibody that will enable detection of both AGO1 and
AGO1x.

We have indeed addressed the specificity of the antibody for immunofluorescence analysis as
the reviewer suggested in Fig. 5b, Fig S2 e,f of our initial submission. Perhaps these figures
were not included in the version of the manuscript that the reviewer downloaded. In the revision,
we replaced Fig. 5b with a panel that shows the intensity of AGO1 and dsRNA signals
separately, to make more clear that the signal from AGO1x is lost in the AGO1x knockout cells.
This is the new Fig 5G. We have also used overexpression of AGO1 / AGO1x from FLAG-
tagged expression constructs (previously Fig S2f and current Fig. EV1C-E) to evaluate whether
the AGO1x staining signal reflected the levels of AGO1x generated from these constructs. We
also included AGO1 staining to illustrate that commercially available AGO1 antibody detects
both isoforms, and that the signal that we obtain with the AGO1x antibody overlaps with the
signal from the canonical AGO1 antibody. Using a pool of siRNAs directed against the AGO1
transcript (which also encodes AGO1x) we found that the signals from both the AGO1x and
AGO1 were strongly depleted. The results we obtained in immunofluorescence were consistent
with those we obtained by WB (previous Fig S2 c,d,g and current Fig EV1E,G).

Figure 3. Here the authors report the use of CRISPR/Cas9 genome editing to eliminate the
AGO1 read-through product (AGO1x). In panel A the authors forgot to indicate the generated
mutations (from their diagnostic genotyping). The authors should also add a methods section
about the detailed characterization of these cell lines. The accompanying western blot seems
confusing as the higher band corresponding to AGO1x is clearly visible in all three lanes using
the pan_AGO1 antibody, but a signal is absent in the two clonal lines using the AGO1x

antibody. The authors should use quantitative imaging technologies like Odyssey or other
methods for this analysis. Cell growth assays should be performed with clonal control cell lines
that arose from the same initial CRISPR/Cas9 transfection. Mutations by repair mistakes of a
guided Cas9 cut generally result in cell clones with variable mutations and deletions that will
have different impact on AGO1 read-through translation. The authors should characterize a set
of cell lines that include functional AGO1 null, AGO1x null, AGO1 null and no effect on AGO1X
expression mutations. This is particularly important as exact homozygous editing events are
rare and MDA-MB-231 are in addition aneuploid (ATCC). Alternatively, the authors could use
precise CRISPR editing (with designed donor constructs) to eliminate either only the canonical
AGO1 STOP codon or only the AGO1x STOP codon. Multiple clonal cell lines have to be tested
for both interference and controls. FLAG-tagged Rescue constructs expressing either FLAG-
AGO1 or FLAG-AGO1x (as used in Fig. S8) could also be used to address the specificity of the
observed effect.

In the methods section and associated tables we have described our CRISPR/Cas9 editing
experiments, including the selection of sgRNAs, the selection and characterization of the cell
lines. We have made this description more verbose in the revised manuscript. We added the
genotyping in Appendix Fig S4, as requested. We have tested a whole number of cell lines, all
of which are clonal but the mutation is not homozygous. As the results we obtained with
different sets of sgRNAs and on two distinct parental lines were very consistent, and different
from those obtained with control lines, in which an unrelated sgRNA set was used (directed
against GFP), we think that our results are not due to spurious effects of transfection or of
CRISPR editing. We could not use the FLAG-tagged constructs to investigate the functional
effects of AGO1x expression because the localization of the encoded protein did not fully
recapitulate the localization of the endogenous form (as discussed in the main text). While we
did attempt to replace the AGO1 stop codon, we did not succeed in doing so. However, we think
that our results with the lines described above address the issue of specificity, and we did not
pursue more complex additional experiments.

Fig 3g: the authors should use fluorescent secondary antibodies and perform a control staining
for normalization.

In Fig. 3g we show classical immunohistochemistry stainings. While we have evaluated the
specificity of the signal (see below) in our initial submission we did not include this in the already
complex Fig. 3.

Ago1x Antibody

IgG Negative Control

Figure 4 and 5 a-f. Here, the authors suggest that depletion of AGO1x results in an increased
interferon response. Like for experiments presented in Figure 3, this set of experiments requires
well established cell lines and control lines. Without thorough characterization of the used cell
lines and appropriate controls these experiments are meaningless.

We point the reviewer to the supplementary material (initial Fig 3, Fig. S4 and S5, current Fig 3
and Appendix Fig S4) that contains the requested characterization. We also explained our
control line in more detail in the text.

For pull-down experiments, the authors should compare AGO1 pull downs with AGO1x pull
downs to identify proteins and RNAs that specifically interact with AGO1x but not AGO1. Using
IgG as a control does not allow discrimination of factors associated with AGO1 or AGO1x.
Because the authors aim to claim a specific function for AGO1x and not a novel function for
overall AGO1, a thorough differential analysis of both isoforms is essential.

As AGO1 and AGO1x differ in their localization, we used the nuclear fraction to increase the
specificity of identification of AGO1x interactors. Furthermore, to discriminate protein interactors

specific for AGO1x or AGO1 we have indeed performed precisely the assay suggested by the
reviewer (former Fig. S7b, current Fig 6E). In the revised manuscript we have carried out
AGO1-IP from the nuclear and cytoplasmic fractions, and found that some putative rRNA
targets of AGO1x are also present in the cytoplasmic AGO1-IP (Appendix Fig S10).

Fig. 5 I and j: Why is DHX9 visible as unspecific background in 2 out of 4 lanes of negative IgG
pull downs (Fig. j) but not in the mass spec analysis (i)?

Mass spectrometry and antibody-based detection have different sensitivities. Therefore, we
used both of these complementary techniques to identify robust interactions and overcome non-
specific interaction of proteins with the beads. While there is some variability in the absolute
signal across samples from a given condition, the average over replicates indicates a consistent
enrichment of DXH9 in the AGO1x pulldown compared to IgG.

Why would PNPT1 knock down increase the abundance of dsRNA in both the parental and the
presumable AGO1x mutant cell lines? Is this effect independent of AGO1x?

dsRNAs are present at a very low though detectable level in the control line (Fig 5F-G, including
new quantification). Thus, if PNPT1 is involved in the removal of dsRNAs as we hypothesize,
we expect that the knockdown of PNPT1 increases the dsRNA signal in the control line, which is
what we see (Fig 6G). Furthermore, the IP experiments that we carried out in response to a
question by reviewer #2 indicate that the interaction of PNPT1 with AGO1x is direct, and not
strictly dependent on RNAs (Appendix Fig S8C).

Minor points:

The authors should revise their introduction to better represent relevant miRNA literature:
Argonaute proteins predominantly localize to the cytoplasm and target mRNAs for degradation.
Nuclear localization and potential function of human Argonaute proteins has long been a
controversial topic in the small RNA field and is not a universally observed phenomenon
(reviewed by David Bartel Cell 2018, PMID: 29570994).

We have mentioned the controversy in the introduction of the revised manuscript.

Referee #2:

Ghosh and co-workers report evidence for the existence of AGO1x, an unexpected Argonaute
isoform, which is proposed to promote proliferation of breast cancer cells by preventing dsRNA-
induced interferon signaling. Key findings include:

- • evidence for expression of AGO1x in human cell lines and breast cancer samples
- • preferential nuclear localization of AGO1x
- • AGO1x promotes cell proliferation
- • AGO1x depletion increases dsRNA accumulation, interferon response, and apoptosis

• AGO1x interacts with PNPT1

This manuscript comes on the heels of Singh et al, 2019, which also demonstrated the
existence of AGO1x and also characterized it's biogenesis and functional properties. Together,
the two studies provide compelling evidence for the existence of AGO1x, and complementary
biological insights that will have a substantial impact on the field.

The manuscript may be improved by addressing the following:

Major Concerns

Conclusions drawn from data in Figs. 1-4 are exciting and compelling. In contrast, the evidence
supporting the final model (as illustrated in Fig. 5i) has several distinct weaknesses:

1) AGO1x and PNPT1 are suggested to physically interact (Figs. 5i-j and S7a-b) and collaborate
to prevent accumulation of 'dsRNA' (Fig. 5l), and yet AGO1x is shown be nuclear (Fig. 2) while
PNPT1 is cytoplasmic (Fig. S7c). How do the authors reconcile this? Is the AGO1x/PNPT1
interaction direct or at least insensitive to RNase A?

We thank the reviewer for this suggestion. Although we could not do the co-staining with the
antibodies for immunofluorescence, as they were both raised in rabbit, we carried out PNPT1
immunoprecipitation from nuclear fractions treated or not with RNase A. The WB of these IPs
indicated that the AGO1x interaction was slightly impaired, but not abrogated by the RNase A
treatment of the lysate prior to the pulldown (Appendix Fig. S8C). This indicates that PNPT1
interacts with AGO1x directly, at least in part. Furthermore the identification of PNPT1 in these
samples suggests that PNPT1 is also present in the nucleus, as also indicated by our initial
imaging analysis (Appendix Fig S8A-B).

2) dsRNA staining in Figs. 5d and 5k is shown in only a few cells. Considering these
observations are used as a major piece evidence for cellular dsRNA up regulation upon Ago1x
knockout, more rigorous analysis is suggested. i.e. what is the distribution of dsRNA staining
levels when surveying >100 cells for each sample?

We appreciate the reviewer for pointing to this observation, which is indeed a major piece of
evidence in our study. We have provided additional quantifications and expanded the
description of these results as suggested Specifically, we counted the numbers of dsRNA foci in
Z stacks of each cell type and provided a new Fig 5F (right panel) showing the statistics in the
revised manuscript. We have further measured the amount of dsRNAs immunoprecipitated from
the different cell types (Fig 5H). Both of these assays show that the total amount of dsRNAs is
~10-fold higher in the mutant lines compared to the control.

3) Purified Argonaute proteins bind RNA non-specifically. Is dsRNA binding (Fig. 5h) specific to
AGO1x? An AGO1 IP could be included as a control to Fig. 5h-since AGO1 levels are >10-fold

AGO1x (Fig. 1d) the AGO1 IP should be dominated by RNAs associated with AGO1 (and not
AGO1x).

The reviewer correctly points out that AGO proteins bind RNA non-specifically. We were aware
of this issue, which we tried to circumvent by purifying the AGO1x protein by IP from the nuclear
fraction before carrying out the qPCR. We reasoned that the presence of AGO1 in the nucleus
is much less pronounced than in the cytoplasm, and thus the signal that we obtained, further
filtered through the AGO1x-IP, should correspond to AGO1x-bound RNAs. To better address
the reviewer's question, in the revised manuscript we also provide qPCRs of the J2 antibody
targets (identified by sequencing) in AGO1-IP of nuclear and cytoplasmic fractions (Appendix
Fig S10). These experiments show that both AGO1 and AGO1x associate with rRNAs that were
identified by sequencing RNAs enriched by the J2 antibody. A precise quantification is difficult,
because the antibody recognizes both AGO1 and AGO1x isoforms (e.g. Fig EV1).

4) Evidence that dsRNA accumulation is responsible for the observed changes in IFN response,
apoptosis, and cell proliferation upon AGO1x knockout is tenuous. Thus, there remains a
substantial possibility that a different mechanism (including altered miRNA function, activation of
the RP-MDM2-p53 pathway, or some other unknown pathway) is the true cause of these
effects. In the absence of direct evidence linking the observed 'dsRNA' to the observed cellular
responses, I strongly suggest downplaying this conclusion.

We thank the reviewer for this comment, which prompted us to investigate more
comprehensively the response of signaling pathways downstream of dsRNAs to AGO1x
depletion. dsRNAs activate PKR both directly and indirectly, via interferon. Our analysis of
transcriptomics data showed that transcriptional targets of IRF1/IRF2/STAT1 are upregulated in
the mutant lines, and CASP3 and 8, which are downstream of these transcription factors, have
increased activity (Fig 4D-E, Appendix Fig S7B,C). By inhibiting JAKs, which are downstream of
IFN signalling, with ruxolitinib, we could rescue the growth of mutant lines (Fig 4G). Finally,
consistent with PKR's inhibition of protein synthesis via eIF2 α phosphorylation (Meurs *et al*,
1992), we found the levels of eIF2 α to be increased in the AGO1x mutant lines (Fig 5D).
Altogether, these experiments provide strong evidence for the dsRNA-IFN axis being
responsible for the observed phenotypes. To determine whether the observed IFN pathway
activation has a functional impact, for the revised manuscript we have also carried out infections
of the HeLa lines with the Sindbis virus and found less accumulation of virus upon infection of
mutant lines compared to the control line, consistent with their higher interferon pathway activity
(Fig EV4). We do not exclude an involvement of the p53 pathway, as crosstalk between PKR
and P53 pathways has been reported (Cuddihy *et al*, 1999) and indeed, GSEA analysis of our
transcriptomics data showed some enrichment of this pathway. However, as the JAK inhibitor
rescues the phenotype, we think that the growth limitation is imposed further upstream, and
hence cannot be solely p53-dependent (Fig 5A). We have discussed these aspects in more
detail in the revised manuscript and also mentioned that we cannot exclude some effect of
miRNAs.

5) Related to the comment above, Fig. 5e shows increased rRNA levels in W1A and W5A cells.
Are ribosomal protein or mature ribosome levels also altered in these cells? Considering the
nucleolar enrichment of AGO1x in Fig. 2b it seems reasonable to suggest a role in ribosome
biogenesis, which if perturbed can initiate nucleolar stress leading to anti-tumor responses.

We thank the reviewer for this comment. Fig. 5e (currently 5L) showed the abundance of rRNA
fragments in the J2 antibody pulldown. We have carried out the suggested evaluation of
ribosome biogenesis, in particular measuring pre- and mature rRNA levels (by qRT-PCR,
relative to GAPDH), and we found them to be reduced in mutant lines (Fig Appendix S11A,B).
Also reduced was the 40S/60S ratio (Appendix Fig S11 C,D). Consistently, the overall
translation was reduced in the AGO1x mutant lines (Appendix S11E,F). The perinucleolar
localization of AGO1x and its binding to rRNAs do suggest a role in ribosome biogenesis, which
could thereby be perturbed in the mutant lines, leading to some degree of nucleolar stress,
consistent with the p53 pathway being identified in our GSEA analysis (Fig 4B and Fig EV3G).
Whether these changes impact anti-tumor responses are indeed an interesting topic for future
work. We have added these points to the discussion.

Minor concerns/suggestions

6) The authors state: "Important for its characterization was the realization that in contrast to the
ectopically-expressed AGO1x, which has cytoplasmic localization both in our and in the study of
Singh et al. (Singh et al , 2019), endogenous AGO1x is found primarily in the nucleus." This
statement does not discuss or acknowledge results in Fig. S4 of Singh et al., 2019, which
appear to show cytoplasmic distribution of endogenous AGO1x.

Indeed, in their Fig S4 Singh et al reported that AGO1x has cytoplasmic distribution. Although a
reanalysis of these data is beyond the scope of our study, we note that the results of Singh et al.
may not be as contradictory to ours as the reviewer thinks (see also our response to the first
question of reviewer #1). Interpreting the image is not straightforward because it is unclear
whether it shows a 3D reconstruction or a single stack and the nucleus is not clearly delineated
by any marker. If we used the distribution of signal intensity to infer nuclear boundaries, we
would also infer some nuclear AGO1x signal in the Singh et al. data as well. Concerning our
own experiments, we found very little bleed-through of AGO1x signal in the cytosol and that only
under high intensity illumination, where the signals were visibly over-saturated as marked by the
software Zen2 (Zeiss). The results were robust, in both MDA-MB-231 and HeLa cell lines.
However, it could also be that difference in localization stems from differences in the cellular
systems, because we used HEK293FT whereas the Singh et al. used HEK293T cells in their
study. For all of these reasons we do not feel that we could expand a lot on the discussion of
the prior data. Nevertheless, we made clear in our Discussion that there are some apparent
discrepancies regarding the localization of AGO1x in different cell lines.

7) Similarly, Singh et al , 2019 report that AGO1x makes up 40% of the endogenous AGO1 in
HEK293 cells, and appears to be similarly expressed in HeLa cells. These results stand in
contrast to the Western blot in Fig. 1d. This difference should be mentioned in some way.

At the reviewer's request, we have mentioned this in our discussion of the results obtained in
the Singh et al. study. We have also added a few lines about prior estimates of endogenous
readthrough frequency in mammalian cells, which have been reported to be at a maximum of
~7% (Loughran *et al*, 2018). Even for a very prominent target of TR, the *gag/pol* transcript of
retroviruses, the probability of readthrough of the *gag* UAG stop codon to form the *gag-pol*
polyprotein necessary for virion assembly is ~5% (Honigman *et al*, 1991)). Therefore, we think
that our estimates, based on the results shown in Fig 1D are more in line with the literature than
estimates of 40% TR efficiency from Singh et al.

8) Where is PNPT1 in Fig. S8a?

We incorporated the PNPT1-IP blot in the revised figure, Fig 6C.

9) Why is the expression change of IFIH1 not shown in Fig. 5b?

IFIH1 was unfortunately not detected in this proteomics experiment and this is why it is not
present in the plot. We mentioned now in the text that the coverage of the proteome was more
limited than that of the transcriptome.

10) Describing rRNA as 'dsRNA' is confusing/inaccurate because, although it does contain a
substantial amount of base-paired helical structures, rRNA is not double stranded.

We tried to correct phrasing accordingly in the revised manuscript.

11) The authors state: "PNPT1, whose siRNA-mediated depletion replicates the AGO1x
depletion phenotype and also leads to the accumulation of dsRNAs." This statement seems
incongruent with data in Fig. 5k, which appears to show more dsRNA staining in the W6A
sample than either siPNPT1 sample.

We agree and have rewritten this statement in the revised manuscript.

12) Fig. S2a: the AGO1x band appears to run much slower than the AGO1 band (in comparison
with Fig. S2b). How did the authors verify that this band corresponds to AGO1x?

The experiment in initial Fig. S2a gave us a hint of AGO1x expression, but we did not use it to
directly confirm the presence of AGO1x, for which we carried out additional analyses (Fig. S2,
current Fig EV1), perturbing the levels of AGO1x and verifying the response of the band
intensity with the specific AGO1x antibody. Regarding the direct comparison of the gels in Figs.
S2a and S2b (current Fig EV1A and B), we note that the gels were run for different times which
could account for the different resolution of the AGO/AGO1x region.

13) It would be reasonable to suggest that the TR extension reduces miRNA affinity (Fig. S8c)
by moving the C-terminal carboxyl group, which interacts with the miRNA 5' phosphate via a
lysine and ordered water molecule in crystal structures of canonical human Ago proteins.

We thank the reviewer for pointing this out. We have added a paragraph of discussion on the
AGO1 structure and possible consequences of a C-terminal extension for binding to miRNAs
and GW182 proteins. The AGO1/AGO1x protein interactomes further support the notion that
AGO1x-GW182 interaction is perturbed.

Referee #3:

In this manuscript Ghosh et al. report that a highly conserved TR isoform of AGO1, AGO1x,
localizes exclusively to the nucleus, in the vicinity of nucleoli, where it prevents accumulation of
double stranded RNAs by an unknown mechanism and thus, supposedly, mitigates the
interferon response and apoptosis. A similar role has also been proposed for Polyribonucleotide
Nucleotidyltransferase 1, a newly identified, specific binding partner of AGO1x. Finally, they also
showed that AGO1x is overrepresented in highly proliferative breast cancer cells and speculate
that targeting AGO1x could potentially represent a new direction for limiting tumor growth.

This is a clearly written, technically well-executed, fairly interesting story supported by numerous
experiments. However, it suffers from 1) insufficient detail (I've had an impression that it was
originally written for some "short-format" journal) forcing the reader to spend quite some time
trying to understand what this or that result means and/or how exactly this or that experiment
was executed, and 2) the lack of any mechanical insight into the presented observations. Below
I summarize all my concerns (in the order of their appearance in the text) that, in my opinion,
should be addressed to improve the quality of this study before its prospective acceptance.

1) I suggest the authors to go through the entire manuscript, a result after result, a figure after
figure (especially Sup. Figures but not only those), and explain in greater detail how each
experiment was done and what can be seen in each figure (expected vs. unexpected and why;
many proteins are listed with no description what they do in cells and why they are expected to
change their expression in this or the other direction, etc.) with a non-expert reader in mind. It
will improve the clarity and readability of your work by a great margin. Also, label the pages and
lines, please.

We apologize for the 'short-format' writing of our manuscript. We have thoroughly revised it to
provide more verbose descriptions of all results, and we hope that the reviewer finds this
version much more understandable. We have also incorporated line and page numbers.

2) Fig. S5a is placed out of order in the main text; in between S2 and S3.

We have now reordered our figures in the revised manuscript to accommodate this change.

3) "Endogenous AGO1x localizes" Fig. 1b should be 2b.

We have rectified the discrepancy in labelling.

4) Fig.3a; the AGO1x strip is smudgy compared to others shown in the work. Since it is a fairly
important result I suggest repeating it aiming for a nicer western blot image.

We thank the reviewer for the suggestion. The smudginess probably stems from the image
conversion from RGB to grayscale. We have adjusted the export parameters to make it more
clear.

5) I am not convinced that the loss of AGO1x results in increased apoptosis. The differences in
Fig. 4e look marginal to me; protein levels of caspases (actually only CASP8 in shown in Fig.
4d) do not dramatically increase either. Other experiments, like cleavage of PARP-1, should be
performed to prove or disprove this point.

We thank the reviewer for this suggestion as well. Our initial inference was based on
transcriptomics data, which indicated increased apoptosis in the AGO1x depleted mutants, as
well as on the imaging of morphological changes in the cells in culture. While realizing that the
timeline of apoptosis and the heterogeneity of cell lines can greatly affect the estimates of
apoptosis, as has been shown earlier (Studzinski, 1999; Härtel *et al*, 2003), we had performed
the Annexin V staining at 16 to 18 hrs post seeding. The numbers that we obtained may also
have been underestimates, as late stage apoptosis cells were not adherent at the time point
chosen for the assay and therefore not collected due to PBS washes prior to trypsinization. We
have now performed a much more thorough characterization of the apoptosis pathway and
included these results in Appendix Fig S7. The results indeed show that the loss of AGO1x
induces apoptosis, with slight variation in the precise pathways involved between MDA-MB-231
and HeLa cell lines. The readout suggested by the reviewer, PARP cleavage relative to the full-
length PARP (Appendix Fig S7D), leads to the same conclusion, that the mutant lines have
higher rates of apoptosis.

6) Figures 4c,d and S6e; please explain how the caspases should behave in your opinion and
why. Why does CASP9 differ between MDA and HeLa lines?

Please see the Appendix Fig S7 and the response to the comment above. While at the pathway
level apoptosis is activated in both lines, there are some detail differences that we cannot
explain at this point.

7) "Hypothesizing that the activation of the interferon response observed upon AGO1x inhibition
is due to the accumulation of double-stranded RNAs (dsRNAs) (Field *et al* , 1967; Lampson *et*
*al* , 1967), we next examined the levels of intracellular dsRNA sensors (Schneider *et al* , 2014).
We found that their expression was indeed increased, both at the mRNA and at the protein
levels (Fig. 5a, b). In addition, using a previously published method (Post *et al* , 2017), we
uncovered evidence for protein phosphorylation events downstream of interferon signaling, in
EIF2AK2 and ADAR (Fig. 5c)." This is a typical example of the "go figure yourself" approach

mentioned above. What are the sensors (how they operate in a few words) and what are not in
the corresp. figure?, why the protein levels of IFIH1 are not shown in 5b?, what "method"?,
phosphorylation of EIF2AK2 and ADAR means what?....

We thank the reviewer for the suggestion to Include a sketch of IFN signaling pathway, which
we now show in Fig 5A. We have expanded the description of the mentioned data and clarified
that some of the proteins were not detected in the proteomic analysis, which generally has less
coverage than an RNA-seq based analysis of the transcriptome.

8) Figure 5d, k - signal quantifications would help.

We have done the quantifications, as requested by both reviewer #2 and #3. PNPT1 KD did not
change the overall relative distribution of the number of foci, only the size of some of the dsRNA
foci. We have therefore plotted the relative distribution of dsRNA foci in the edited and control
cell lines (Fig 5F).

9) The fact that the source of dsRNA lies predominantly, if not solely (based on Fig. 5e), in
rRNA is rather unexpected. The authors propose that: "...the increased demand for ribosome
biogenesis in rapidly dividing cells requires the readthrough AGO1x isoform for resolving dsRNA
structures in rRNAs and other molecules."; also rather unexpected, but why not. Since no other
explanation has been offered, either for the AGO1x role in dsRNA metabolism in the nucleus, or
for its contact with PNPT1, I suggest the authors to explore this idea in greater detail. If correct,
ribosome biogenesis should be affected in AGO1x null (as well as PNPT1 knocked-down) cells,
which can be easily tested with a bunch of well-established approaches.

We have carried out additional experiments, as suggested also by reviewer #2. Specifically, we
have quantified rRNA levels, the relative abundance of 40 and 60S ribosomal subunits and the
overall translation, all of which were reduced in the mutant lines (Appendix Fig S11). We think
that the results support a role of AGO1x on ribosome biogenesis. In addition, we showed that
AGO1x is co-IPed with PNPT1 in control cells (Fig 6). The statement to which the reviewer
refers is in the discussion of our manuscript, where we thought we can offer some speculation.

10) What could be reason for the observed differences between endogenously vs. ectopically
expressed AGO1x? Any idea?

We were puzzled by these differences as well. We have started to investigate a few
possibilities, in particular the effect of the amino acid that is incorporated at the stop codon, but
at this point we cannot provide a conclusive answer.

References

Cuddihy AR, Li S, Tam NWN, Wong AH-T, Taya Y, Abraham N, Bell JC & Koromilas AE (1999)

- Double-Stranded-RNA-Activated Protein Kinase PKR Enhances Transcriptional Activation
by Tumor Suppressor p53. *Molecular and Cellular Biology* **19**: 2475–2484 Available at:
<http://dx.doi.org/10.1128/mcb.19.4.2475>
- Härtel S, Zorn-Kruppa M, Tykhonova S, Alajuuma P, Engelke M & Diehl HA (2003)
Staurosporine-induced apoptosis in human cornea epithelial cells in vitro. *Cytometry Part A*
**55A**: 15–23 Available at: <http://dx.doi.org/10.1002/cyto.a.10068>
- Honigman A, Wolf D, Yaish S, Falk H & Panet A (1991) cis Acting RNA sequences control the
gag-pol translation readthrough in murine leukemia virus. *Virology* **183**: 313–319
- Loughran G, Jungreis I, Tzani I, Power M, Dmitriev RI, Ivanov IP, Kellis M & Atkins JF (2018)
Stop codon readthrough generates a C-terminally extended variant of the human vitamin D
receptor with reduced calcitriol response. *J. Biol. Chem.* **293**: 4434–4444
- Meurs EF, Watanabe Y, Kadereit S, Barber GN, Katze MG, Chong K, Williams BR &
Hovanessian AG (1992) Constitutive expression of human double-stranded RNA-activated
p68 kinase in murine cells mediates phosphorylation of eukaryotic initiation factor 2 and
partial resistance to encephalomyocarditis virus growth. *J. Virol.* **66**: 5805–5814
- Studzinski GP (1999) Apoptosis: A Practical Approach Oxford University Press, USA

Prof. Mihaela Zavolan
University of Basel
Biozentrum
Klingelbergstrasse 50/70
Basel, Basel-Stadt CH-4056
Switzerland

14th Apr 2020

Re: EMBOJ-2019-103922R
AGO1x prevents dsRNA-induced interferon signaling to promote proliferation of breast cancer cells

Dear Mihaela,

Thank you for submitting your revised manuscript for our consideration. It has now been seen once more by the three original referees (see comments below). As you will see, the referees acknowledge that you have performed additional experiments and added clarifications, but are also not yet convinced that the conclusions are sufficiently supported by the data. Given the overall interest in the study, we would like to give you the opportunity to address the remaining issues in an exceptional second round of revision.

Referee #1 continues to have major concerns regarding the heterozygous cell lines and the levels of AGO1x. While the other referees agree that homozygous cell lines would be a more rigorous approach, they find that decisively showing a substantial depletion of AGO1x in the cell lines currently used would significantly contribute to addressing referee #1's concerns and supporting the conclusions drawn in Figures 4- 6. Both referees are not convinced by the Western blot provided in Figure 3A, in particular by the reformatting of the blot rather than its repetition (ref #3 ad 4; ref#2 cross-commenting). To consider this study further for publication, this issue must be adequately addressed and convincing evidence for the reduced levels of AGO1x provided, as this affects all main conclusions of the following figures. Referee #2 suggests doing so by performing replicates (at least 3, preferably more) of the Western blots and quantitating AGO1x and AGO1 levels relative to the GAPDH control to provide clean and quantitative documentation of significant depletion of AGO1x (without effects on AGO1). Moreover, both referee #2 and #3 are not fully convinced of the small differences between wild type and the mutant cell lines in figure S11. The specific questions referee #3 raises should thus be addressed and the text carefully revised to more accurately interpret the results.

As indicated in the previous letter, it is normally EMBO Journal's policy to allow only one round of major revision, such that it is now crucial that you address the major concerns regarding Figure 3 fully and adequately respond to the other remaining issues raised. If you have any questions regarding this revision or would like to discuss how to proceed in more detail, please feel free to contact me.

Kind regards,

Stefanie Boehm

Stefanie Boehm
Editor
The EMBO Journal

Referee #1:

main concern: Fig. 3: the authors claim to have generated a genetic mutation in the AGO1 gene that eliminates the read-through product (AGO1x) without showing sufficient evidence that their cell lines indeed have specifically eliminated AGO1x. First, the authors add a supplementary figure (Fig S4) that shows that the mutations affect different regions in the 3'UTR/read through region without any clear indication how the new read-through protein looks like. Second, their seq track suggest and they also directly state that their mutations are heterozygous, meaning that the second AGO1 allele is still wild type and can produce the read-through product AGO1X. Third, the authors state in the rebuttal that they attempted the rescue of a potential AGO1x null phenotype but failed to do so. Their own data and words raise additional concerns not only about the experiments themselves but also about their general interpretation of data. Their model remains unsupported by the presented data. Their data remain insufficiently controlled and explained. Their observations do not support their interpretations. I do not recommend this manuscript for publication.

Referee #2:

The authors have addressed my previous comments with thought and care. I congratulate them on an exciting and compelling study.

Referee #3:

This manuscript has greatly improved, no doubt about that. However, I think that a few more issues shown below have to be resolved before it can be published.

Ad 4) I am a bit worried by this response - a different image processing showing now what should be seen? It is relatively very easy to repeat this experiment to convince the reader that everything has been done properly, isn't it? I do not doubt it, I am just somewhat surprised.

Ad 7) Greatly improved! Here, however, Fig. 5D is technically wrongly executed. In addition to what you show now, you must also show the eIF2alpha protein levels and normalize your "alpha-P" signal to them to be 100% sure about your claims.

9) Thank you for your effort but I am not convinced by these data at all.

S11A - B: typically, a defect in ribo biogenesis results in increased levels of precursors and decreased levels of mature forms of rRNAs. In your case everything is down, which would suggest that the biogenesis pre se is okay, but the rDNA transcription is significantly mitigated. Can you test it?

S11C - D: Perfectly fits with the above; visually, I see no significant differences whatsoever - how many times did you repeat this experiment (I see no error bars in the panel D)? At least three

times would be required for such a marginal difference, if any. Also, the wt (blue) tracing looks weird and prevents you from making any conclusions, in my opinion. Finally, no CHX should be added to the buffer and no Mg ions as well (this is much better than supplying the buffer with excessive amounts of EDTA).

S11F: Same as above, multiple repetitions should be carried out and the Polysome to Monosome (P/M) ratios with SD should be calculated before any conclusions can be drawn. There could be a slight P/M decrease in your mutants, I agree - perhaps due to an overall reduction in the production of ribosomes but not a defect in their biogenesis, you just have to convincingly demonstrate it.

Referee #1:

main concern: Fig. 3: the authors claim to have generated a genetic mutation in the AGO1 gene that
eliminates the read-through product (AGO1x) without showing sufficient evidence that their cell lines
indeed have specifically eliminated AGO1x. First, the authors add a supplementary figure (Fig S4) that
shows that the mutations affect different regions in the 3'UTR/read through region without any clear
indication how the new read-through protein looks like. Second, their seq track suggest and they also
directly state that their mutations are heterozygous, meaning that the second AGO1 allele is still wild type
and can produce the read-through product AGO1X. Third, the authors state in the rebuttal that they
attempted the rescue of a potential AGO1x null phenotype but failed to do so. Their own data and words
raise additional concerns not only about the experiments themselves but also about their general
interpretation of data. Their model remains unsupported by the presented data. Their data remain
insufficiently controlled and explained. Their observations do not support their interpretations. I do not
recommend this manuscript for publication.

We did not claim that we have *eliminated* AGO1x, but rather that we depleted it. MDA-MB-231 cells
are aneuploid, 'with chromosome counts in the near-triploid range'
(https://genome.ucsc.edu/ENCODE/protocols/cell/human/MDA-MB-231_Struhl_protocol.pdf) and
achieving appropriate editing of all three chromosomes would be difficult. The editing was demonstrated
both by sequencing the DNA loci as well as by aligning the RNA-seq reads from these lines to the
corresponding locus (Appendix Fig. S4A,C). The depletion of the protein was apparent both in the western
blots (Fig. 3A lower panel) and in imaging data (Fig. 5G). Nevertheless, to hopefully dispel any remaining
concerns, we have carried out another round of western blotting experiments, with three replicates for
each of the control, W1A and W6A mutant MDA-MB-231 lines. The results re-confirm the depletion of
AGO1x in the mutant lines with respect to control, while the total AGO1 levels were not changed
substantially, likely because the abundance of AGO1 is higher than that of AGO1x. These results are shown
in revised Fig. 3A, lower panel. We have further repeated the western blots in longer running gels (where
lower MW proteins run out of the gel), as done by (Singh *et al*, 2019). Here we found that the faint, higher
29 MW band identified by the AGO1 antibody in control cells is reduced in intensity in the mutant lines. From
30 these better resolved gels we have again quantified the canonical AGO1 band and by probing the gels
with the AGO1x antibody we specifically quantified AGO1x. The results confirm the depletion of AGO1x
but not AGO1 in the mutant lines relative to control (Appendix Fig. S4E, quantification in S4F).

Regarding the rescue, we noted that the reason for not pursuing assays with the construct for ectopically-
expressing the tagged protein was that this protein had a different localization than the endogenous
AGO1x.

We did not further attempt to determine whether the mutant lines express yet another version of a
readthrough protein. In the study, we generated 4 distinct mutant lines (2 in MDA-MB-231 and 2 in HeLa)
by distinct editing events. All these lines have in common a reduced expression of AGO1x, along with
similar molecular makeup and phenotypes. While some side effect may be present in one cell line or the

other, this being the same in the 4 distinct lines is unlikely. The most direct and parsimonious explanation
is that the phenotypes are due to what is common to all lines, namely the depletion of AGO1x.

Referee #2:

The authors have addressed my previous comments with thought and care. I congratulate them on an
exciting and compelling study.

We thank the reviewer for explicitly acknowledging our efforts to address their comments.

Referee #3:

This manuscript has greatly improved, no doubt about that. However, I think that a few more issues shown
below have to be resolved before it can be published.

Ad 4) I am a bit worried by this response - a different image processing showing now what should be seen?
It is relatively very easy to repeat this experiment to convince the reader that everything has been done
properly, isn't it? I do not doubt it, I am just somewhat surprised.

In our initial revision, we responded to the reviewer's comment about the smudginess of the band and
did not realize that the reviewer expected us to redo the experiment and replace the figure. Indeed, this
is not a problem. We have now carried out western blot analysis of AGO1 and AGO1x in triplicate, using
the respective antibodies. We have incorporated these new results in our revised Fig. 3A. As the AGO1
antibody identified a single band in these blots, we also repeated the experiment running the gel longer
(as shown earlier in our manuscript and also done in the study of (Singh *et al*, 2019); this caused low MW
proteins to run out of the gel). We probed these gels first with AGO1 antibody, which revealed not only
the band corresponding to the canonical AGO1, but also the faint, higher MW band, present in the control
line and reduced in intensity in the mutant lines. By further probing the blot with AGO1x antibody, we
confirmed the AGO1x depletion. We have inserted these results in the Appendix Fig S4E-F. We hope that
all of these blots dispel any doubt that AGO1x is depleted in the mutant lines.

Ad 7) Greatly improved! Here, however, Fig. 5D is technically wrongly executed. In addition to what you
show now, you must also show the eIF2 α protein levels and normalize your "alpha-P" signal to them
to be 100% sure about your claims.

Our proteomics data showed that eIF2 α did not change significantly in the mutant lines, and if anything,
the small (< 20%) change was in the direction of decrease in mutant compared to control lines (log₂ ratio
to control was -0.28 in the W1A line and -0.16 in the W6A line). In contrast, the WBs of p-eIF2 α indicated
that this form was *increased* in the mutant lines relative to GAPDH control. However, to address the
reviewer's comment, we have augmented Fig. 5D with the results of probing the same cell lysates for eIF2 α .

We have additionally carried out a parallel quantification of eIF2 α and p-eIF2 α in triplicate, by western
blotting from samples used earlier in Supplementary Fig S7D. We have included these additional replicates
and their quantification in Appendix Fig S7D. These results confirm that eIF2 α does not change
significantly in the mutant lines, while p-eIF2 α increases 1.5-2-fold.

9) Thank you for your effort but I am not convinced by these data at all.

S11A - B: typically, a defect in rRNA biogenesis results in increased levels of precursors and decreased levels
of mature forms of rRNAs. In your case everything is down, which would suggest that the biogenesis pro-
cess is okay, but the rDNA transcription is significantly mitigated. Can you test it?

Perhaps our use of the term 'rRNA biogenesis' was less specific than the reviewer's, as we meant the
entire chain of rRNA production and processing. The assays that we carried out tried to cover multiple
aspects of the process, as the reviewer asked. In particular, our qRT-PCR results (with three independent
samples per condition) indicated that both pre-rRNA and mature rRNA levels are reduced in the mutant
lines (Appendix Fig. S11A-B). This is certainly consistent with reduced transcription, which we have tried
to clarify by updating the discussion. The reduction in rRNA levels is reflected in a decrease in total
translation in the mutant lines, which we have shown by the incorporation of the methionine analogue L-
homopropargylglycine (HPG) (Su Hui Teo *et al*, 2016; Essig *et al*, 2017) in two independent replicate
experiments in HeLa lines and now also in three biological replicates in MDA-MB-231 lines (Appendix Fig.
S11E-F). Further dissecting which step of ribosome subunit production or assembly is perturbed would
require in-depth testing of multiple possibilities that we think would constitute a separate study. We hope
that the reviewer agrees.

S11C - D: Perfectly fits with the above; visually, I see no significant differences whatsoever - how many
108 times did you repeat this experiment (I see no error bars in the panel D)? At least three times would be
required for such a marginal difference, if any. Also, the wt (blue) tracing looks weird and prevents you
from making any conclusions, in my opinion. Finally, no CHX should be added to the buffer and no Mg ions
as well (this is much better than supplying the buffer with excessive amounts of EDTA).

For that figure, we have assessed the 40S/60S ratio in duplicate and we showed the results for individual
replicates (which is why there were no error bars). To convince the reviewer about this particular aspect,
we added two additional replicates per cell line, with the quantification of all four replicates shown in
Appendix Fig. S11D. A representative run is also shown in Appendix Fig. S11C. For each replicate, we
assayed one sample of control, one of W1A and one of W6A mutant lines, and we computed the 40S/60S
ratios in mutants relative to the corresponding control. We hope that these dispel any remaining doubts
that the stoichiometry of 40S/60S subunits is affected in the mutant lines. Regarding the protocol for the
estimation of 40S/60S ratio, the one we employed is widely used (Fleischer *et al*, 2006; Koplín *et al*, 2010;
Santos *et al*, 2011; Hansji *et al*, 2016) and, especially with the current limitations on experimental work,
we opted to do the additional analysis as before rather than trying to implement other protocols.

S11F: Same as above, multiple repetitions should be carried out and the Polysome to Monosome (P/M)
ratios with SD should be calculated before any conclusions can be drawn. There could be a slight P/M

decrease in your mutants, I agree - perhaps due to an overall reduction in the production of ribosomes
but not a defect in their biogenesis, you just have to convincingly demonstrate it.

The reviewer's comment seems to highlight that the P/M ratio is a rather qualitative readout of
translation. Therefore, rather than adding additional replicates of these profiles, we decided to pursue a
more quantitative readout, based on HPG incorporation. We have carried out this experiment in triplicate
and found a clear and reproducible reduction in translation in the mutant MDA-MB-231 lines compared
to control. We have added these data as Appendix Fig. S11F, complementing the previous analysis of HeLa
lines.

**References**

Essig K, Hu D, Guimaraes JC, Alterauge D, Edelmann S, Raj T, Kranich J, Behrens G, Heiseke A, Floess S,
Klein J, Maiser A, Marschall S, Hrabě de Angelis M, Leonhardt H, Calkhoven CF, Noessner E, Brocker
139 T, Huehn J, Krug AB, et al (2017) Roquin Suppresses the PI3K-mTOR Signaling Pathway to Inhibit T
Helper Cell Differentiation and Conversion of Treg to Tfr Cells. *Immunity* **47**: 1067–1082.e12

Fleischer TC, Weaver CM, McAfee KJ, Jennings JL & Link AJ (2006) Systematic identification and
functional screens of uncharacterized proteins associated with eukaryotic ribosomal complexes.
*Genes Dev.* **20**: 1294–1307

Hansji H, Leung EY, Baguley BC, Finlay GJ, Cameron-Smith D, Figueiredo VC & Askarian-Amiri ME (2016)
ZFAS1: a long noncoding RNA associated with ribosomes in breast cancer cells. *Biol. Direct* **11**: 62

Koplín A, Preissler S, Ilina Y, Koch M, Scior A, Erhardt M & Deuerling E (2010) A dual function for
chaperones SSB-RAC and the NAC nascent polypeptide-associated complex on ribosomes. *J. Cell*
*Biol.* **189**: 57–68

Santos MCT, Goldfeder MB, Zanchin NIT & Oliveira CC (2011) The essential nucleolar yeast protein
Nop8p controls the exosome function during 60S ribosomal subunit maturation. *PLoS One* **6**:
e21686

Singh A, Manjunath LE, Kundu P, Sahoo S, Das A, Suma HR, Fox PL & Eswarappa SM (2019) Let-7a-
regulated translational readthrough of mammalian AGO1 generates a microRNA pathway inhibitor.
*EMBO J.* **38**: e100727

Su Hui Teo C, Serwa RA & O'Hare P (2016) Spatial and Temporal Resolution of Global Protein Synthesis
during HSV Infection Using Bioorthogonal Precursors and Click Chemistry. *PLoS Pathog.* **12**:
e1005927

Prof. Mihaela Zavolan
University of Basel
Biozentrum
Klingelbergstrasse 50/70
Basel, Basel-Stadt CH-4056
Switzerland

23rd Jun 2020

Re: EMBOJ-2019-103922R1

AGO1x prevents dsRNA-induced interferon signaling to promote proliferation of breast cancer cells

Dear Mihaela,

Thank you for submitting the revised manuscript and addressing the remaining referee concerns. We have now received the reports from two of the original referees (see comments below). I am pleased to say that they overall find that their comments have been satisfactorily addressed and now support publication. Referee #3 has some minor issues left that can be incorporated into the final revised version. In addition, I would like to also ask you to address a number of editorial issues that are listed in detail below. Please make any changes to the manuscript text in the attached document only using the "track changes" option. Once these remaining issues are resolved, we will be happy to formally accept the manuscript for publication.

Thank you again for giving us the chance to consider your manuscript for The EMBO Journal. I look forward to receiving your final revision. Please feel free to contact me if you have further questions regarding the revision or any of the specific points listed below.

Kind regards,

Stefanie

Stefanie Boehm
Editor
The EMBO Journal

Referee #2:

In this second revision, Ghosh and Guimaraes and co-workers included new Western blot data that more carefully determines the extent of AGO1x knock-out in the cell lines used in their experiments. The data show knock-out, using two different sgRNAs, is reproducible and statistically significant. This evidence strengthens the weight of conclusions drawn in subsequent sections of the manuscript substantially.

Referee #3:

I applaud the authors for the great work! I have only a few very minor comments.

Ad 4) This response is convincing for me now.

Ad 7) This response is convincing for me now.

Ad 7) I am satisfied with the way the data were processed, as well as with the text edits. Perhaps, I would only modify this sentence as follows: "This reduction is consistent with AGO1x impacting the pre-rRNAs, but whether THE rDNA transcription or ITS post-transcriptional processing steps ARE AFFECTED remains to be clarified."Also, labeling of X-axis in S11E seems to be cropped.

And no, the P/M ratio is NOT a rather qualitative readout of translation. In fact, there is no doubt that correctly executed polysome profiling is a well of treasures for translational studies that can be perfectly qualitative. No irony needed here, check the literature, it is overwhelmed with conclusions based on reproducible and well-controlled P/M ratios.

BIOZENTRUM
Universität Basel
The Center for
Molecular Life Sciences

Mihaela Zavolan
Biozentrum &
Swiss Institute of Bioinformatics
Klingelbergstrasse 50 – 70
CH-4056 Basel

Swiss Institute of
Bioinformatics

Dear Stefanie

We are indeed very happy for the very positive comments of the reviewers. We have answered the editorial questions in the manuscript and uploaded the figures as requested.

We very much look forward to see our manuscript in print.

Best wishes

Mihaela Zavolan

Prof. Mihaela Zavolan
University of Basel
Biozentrum
Klingelbergstrasse 50/70
Basel, Basel-Stadt CH-4056
Switzerland

7th Jul 2020

Re: EMBOJ-2019-103922R2
Prevention of dsRNA-induced interferon signaling by AGO1x is linked to breast cancer cell proliferation

Dear Mihaela,

Thank you again for submitting the final revised version of your manuscript. I am pleased to inform you that we have now accepted it for publication in The EMBO Journal.

Your article will be processed for publication in The EMBO Journal by EMBO Press and Wiley, who will contact you with further information regarding production/publication procedures and license requirements.

Should you be planning a Press Release on your article, please get in contact with embojournal@wiley.com as early as possible, in order to coordinate publication and release dates.

Congratulations on your successful publication, and thank you again for this contribution to The EMBO Journal! Please continue to consider EMBO Journal for your work in the future.

Kind regards,

Stefanie

Stefanie Boehm
Editor
The EMBO Journal

Author Checklist

EMBO PRESS

YOU MUST COMPLETE ALL CELLS WITH A PINK BACKGROUND. PLEASE NOTE THAT THIS CHECKLIST WILL BE DIRTY WHEN ALONGSIDE YOUR DATED

Corresponding Author Name: **Mihaela Zavolan, Joao Guimaraes**
Journal Submitted to: **EMBO**
Manuscript Number: **EMBOJ-2019-103922R2**